# Heat Shock Transcription Factor CgHSF1 Is Required for Melanin Biosynthesis, Appressorium Formation, and Pathogenicity in *Colletotrichum gloeosporioides*

**DOI:** 10.3390/jof8020175

**Published:** 2022-02-11

**Authors:** Xuesheng Gao, Qiannan Wang, Qingdeng Feng, Bei Zhang, Chaozu He, Hongli Luo, Bang An

**Affiliations:** 1Hainan Key Laboratory for Sustainable Utilization of Tropical Bioresource, College of Tropical Crops, Hainan University, Haikou 570228, China; 19095131210013@hainanu.edu.cn (X.G.); wangqiannan@hainanu.edu.cn (Q.W.); fqd2429009287@163.com (Q.F.); zhangbei@hainanu.edu.cn (B.Z.); czhe@hainanu.edu.cn (C.H.); 2Sanya Nanfan Research Institute, Hainan University, Sanya 572025, China

**Keywords:** HSF1, melanin, appressorium, pathogenicity, *Colletotrichum gloeosporioides*

## Abstract

Heat shock transcription factors (HSFs) are a family of transcription regulators. Although HSFs’ functions in controlling the transcription of the molecular chaperone heat shock proteins and resistance to stresses are well established, their effects on the pathogenicity of plant pathogenic fungi remain unknown. In this study, we analyze the role of CgHSF1 in the pathogenicity of *Colletotrichum gloeosporioides* and investigate the underlying mechanism. Failure to generate the *Cghsf1* knock-out mutant suggested that the gene is essential for the viability of the fungus. Then, genetic depletion of the *Cghsf1* was achieved by inserting the repressive promoter of nitrite reductase gene (PniiA) before its coding sequence. The mutant showed significantly decrease in the pathogenicity repression of appressorium formation, and severe defects in melanin biosynthesis. Moreover, four melanin synthetic genes were identified as direct targets of CgHSF1. Taken together, this work highlights the role of CgHSF1 in fungal pathogenicity via the transcriptional activation of melanin biosynthesis. Our study extends the understanding of fungal HSF1 proteins, especially their involvement in pathogenicity.

## 1. Introduction

The genus *Colletotrichum* spp. is a group of plant pathogenic fungi that can infect a wide range of plants worldwide [1]. There is a total of 600 *Colletotrichum* species identified by now; among these species, *Colletotrichum gloeosporioides* can infect over 470 plant species and cause anthracnose diseases in both aerial plant parts and in post-harvest fruits, such as banana, mango, avocado, coffee [2].

Most of *Colletotrichum* species employ a hemibiotrophic lifestyle, including conidia germination, melanized appressorium formation, biotrophic intracellular hyphae, and a necrotrophic phase on dead host cells [1]. During infection processes, *Colletotrichum higginsianum*, *Colletotrichum orbiculare*, and *C. gloeosporioides* could adjust their strategies to switch between these stages by regulating the expression of virulence factors, such as effectors, secondary metabolite synthesis enzymes, and degradative enzymes [3,4]. In addition, a number of transcription factors (TFs) were found to play important roles in controlling the switch from non-pathogenic parasite to pathogeny in pathogenic fungi. For example, Wor1, a conserved fungal TF from *Candida albicans*, controls the white–opaque switching of cells and is required for virulence to humans [5,6]. In plant pathogenic fungi from the genus *Fusarium*, SGE1 proteins, the orthologs of Wor1, regulate effector genes and secondary metabolite gene clusters [7,8,9,10], and are required for parasitic growth [11]. In *Botrytis cinerea*, *Verticillium dahliae*, *Zymoseptoria tritici*, *Ustilago maydis*, and *Magnaporthe oryzae*, orthologs of Wor1 also participate in the regulation of the expression of effector genes and virulence to the hosts [12,13,14,15,16]. Some other TFs, such as the forkhead transcription factor Fox1 and zinc finger transcription factor Mzr1 of *Ustilago maydis*, are involved in the regulation of effector genes [17,18].

Heat shock transcription factors (HSFs) are a family of transcription regulators, which were thus named for they can regulate the expression of heat shock proteins (HSPs) in cells [19,20]. Since the first discovery and being cloned in *Drosophila* and yeast [21,22], HSFs have been identified in a variety of species. HSFs are conserved in eukaryotes from fungi, such as *Saccharomyces cerevisiae*, which has only one HSF gene, to humans whose genome encodes six HSF proteins [22,23]. In plants, HSFs are a large and diverse gene family with more complex functions due to their great differences in size and sequence [24]. HSFs were originally described to recognize and specifically bind the conserved motif of the heat shock element (HSE) in the promoter region of *HSP* genes, thereby increasing biological resistance to heat stress, chemical stimuli, and oxidative stress [24,25,26]. In yeast, HSF1 is essential for survival and the modulation of chaperone levels in response to growth temperature [27]. In mouse, the HSF1-dependent expression of HSP chaperones is required to maintain redox homeostasis and to regulate antioxidative defenses [28]. Additionally, the HSFs of plants play roles in response to oxidative, heavy metals, and biotic stresses during the development of and differentiation in plants [29,30]. In addition to the roles in stress response, there are growing evidences indicating that HSFs play pivotal roles in development and differentiation [24,31]. For example, HSFB2a is required for gametophytic development in *Arabidopsis* [32]; the mouse HSF4 regulates the development of lenses [33]; especially in *C. albicans*, both the depletion and overexpression of HSF1 could induce the transition from yeast to filamentous growth, highlighting the key role of HSF1 in virulence [34]. However, whether HSFs are required for development and virulence of plant pathogenic fungi is still elusive.

In the present study, our goal is to investigate whether CgHSF1 is required for the pathogenicity of *C. gloeosporioides*, and to figure out the possible mechanism. We found that *CgHSF1* depletion impairs the expression of melanin biosynthesis genes, compromises appressorium formation, and leads to the decrease in pathogenicity. This work provides new insight into HSF1′s role in fungal pathogenicity.

## 2. Materials and Methods

### 2.1. Fungal Strains and Plant Material

*C. gloeosporioides Hevea* (BioSample: SAMN17266943 (https://www.ncbi.nlm.nih.gov/biosample/17266943 (accessed on 9 January 2021)) was isolated from *H. brasiliensis* previously and kept on potato dextrose agar medium (PDA) at 28 °C. The *H. brasiliensis* cultivar Reyan 7-33-97 was cultured in a glass house and used for the pathogenicity assay.

### 2.2. Bioinformatics Analysis

The amino acid sequence of CgHSF1 was searched for the conserved domains against Pfam database. The HSF1 homologues from other phytopathogenic fungi were retrieved from NCBI GenBank database, and the maximum likelihood tree was constructed using JTT + G+I model and 1000 bootstrap with MEGA 11 [35].

### 2.3. Construction of the Gene Knock-Out Strain, PniiA-Cghsf1 Inducible Strain, and the GFP or FLAG Tagged Strains

The knock-out of *Cghsf1* was attempted by replacing the *Cghsf1* gene with the acetolactate synthase gene (SUR) cassette from *M. oryzae*, which confers resistance to chlorimuron ethyl (a sulfonylurea herbicide). Briefly, the upstream and downstream flanking fragments of *Cghsf1* were amplified from genomic DNA with TransStart^®^ FastPfu DNA Polymerase (TransGen Biotech, Beijing, China) and cloned into the plasmid pBS-SUR [36]. Then, the flanking sequences together with the truncated SUR were amplified and used for protoplast transform (Appendix A).

To modulate the expression of *Cghsf1*, an inducible promoter (PniiA) of nitrite reductase gene (niiA) was inserted between the coding sequence of *Cghsf1* and its native promoter. The hygromycin B resistance cassette (HPT) was used as the selective marker. Firstly, the upstream flanking fragments of *Cghsf1* were amplified with primer pairs HSF-5F/HSFhpt-5MR, with the reverse primer having 17 nucleotides (nts) complementary to HPT sequence; and a truncated HPT fragment was amplified with primer pairs HSFhpt-5MF/hpt-SPLR, with the forward primer having 17 nts complementary to *Cghsf1*; then the two fragments were ligated together via overlap PCR with the two fragments as templates. Secondly, the other truncated HPT (with primers hpt-SPLF/hptniiA-MR), the PniiA promoter (with hptniiA-MF/niiAHSF-MR), and the 800 bp nucleotides of 5′ part of *Cghsf1* (*Cghsf1*-L) sequence (with niiAHSF-MF/HSF-L-R) were amplified, and then the three fragments were fused by PCR. After that, the two fused fragments were used for protoplast transformation. 

The modified plasmid pMD19-T, which contains the GFP/FLAG coding sequence, a terminator of the tryptophan synthase of *Aspergillus nidulans* (TtrpC), and the HPT cassette, was used to tag the CgHSF1 with GFP or FLAG. The 700 bp nucleotides of the 3′ part of *Cghsf1* without stop codon (*Cghsf1*-R) were amplified with the primers HSF-R-F/HSF-R, and cloned into the plasmid right before the GFP/FLAG tag; then the fragment containing the 3′ part of *Cghsf1*, GFP/FLAG, TtrpC, and the truncated HPT was prepared by PCR using the primers HSF-R-F/hpt-SPLR. The downstream flanking fragments of *Cghsf1* were amplified with hptHSF3-MF/HSF-3R, and fused with the other part truncated HPT (amplified with primers hpt-SPLF/hptHSF3-MR) using PCR. After that, the two fused fragments were used for protoplast transform (Appendix A).

Protoplast preparation and transformation were conducted as described previously [36]. The transformants were selected by resistance to chlorimuron ethyl or hygromycin B. To verify the correct integration of the recombinant fragments into the target site of the genome, two independent PCR diagnoses were conducted with the primer pairs HSF-DF/hpt-DR and niiA-DF/HSF-L-DR, with one primer being located outside of the homologous fragments and one being located inside of the expression cassette for each primer pair. In addition, an additional PCR was conducted to confirm the correctness of the inducible expression cassette, with one primer complementary to the beginning of PniiA and the other primer complementary to the down-stream sequence of *Cghsf1*. Besides, the full length of *Cghsf1* in the GFP/FLAG tagged strains were amplified and sequenced for correctness. After that, the heterokaryon of the correct transformants were purified by single conidia isolation. The primers used are listed in Appendix A.

### 2.4. Pathogenicity Assay

The pathogenicity assay was performed as described in our previous report [36]. *C. gloeosporioides* conidia were collected and resuspended in 0.5% (*w/v*) malt extract broth (Difco, Thermo Fisher, Waltham, MA, USA) to a final concentration of 2 × 10^5^ conidia mL^−1^. The intact “light green” leaves were detached from a rubber tree and used for the pathogen inoculation. The leaves were divided into two groups, with one group of intact leaves and the other group of leaves being pre-wounded with a sterile needle. Then, 5 μL of droplets of the conidial suspensions were inoculated onto the leaves. After that, the inoculated leaves were kept in Petri dishes with lids at 28 °C under natural illumination for 3 d, and the disease symptoms were scored. Each treatment contained three replicates of 10 leaves, and the entire experiment was repeated 3 times.

### 2.5. Quantitative RT-PCR Analysis

For the analysis of the expression levels of *CgniiA*, *CgniaD*, and *Cghsf1* in response to different nitrogen sources, *C. gloeosporioides* WT strain were cultured on a minimal medium (0.5 g L^−1^ KCl, 1 g L^−1^ KH_2_PO_4_, 0.5 g L^−1^ MgSO_4_·7H_2_O, 0.01 g L^−1^ FeSO_4_.7H_2_O, 20 g L^−1^ sucrose, pH 6.9) supplemented with NaNO_3_ (20 mmol L^−1^), ammonium tartrate (C_4_H_12_N_2_O_6_) (10 mmol L^−1^), and glutamine (10 mmol L^−1^) as nitrogen sources. The expression levels of *Cghsf1* in the repressive mutant strain during in vitro and in planta stages were also assayed. For the collection of in vitro samples, conidia were inoculated into the liquid minimal medium supplemented with different nitrogen sources at a concentration of 10^6^ conidia mL^−1^; after incubation at 160 rpm, 28 °C for 1 d, the mycelium was collected and used for RNA extraction. For the collection of in planta samples, the strains were inoculated onto pre-wounded leaves as mentioned above. After incubation at 28 °C for 2 d, the lesion area in the leaves were gathered and used for RNA extraction. The samples were disrupted in liquid nitrogen using a mortar with a pestle, and the RNA was extracted according to the manufacture’s instruction by using the RNAprep Pure Plant Plus Kit (TIANGEN Biotech, Beijing, China). First strand cDNA was synthesized with FastKing gDNA Dispelling RT SuperMix (TIANGEN Biotech, Beijing, China). Then, a quantitative RT-PCR analysis was performed with ChamQ SYBR Color qPCR Master Mix (Vazyme, Nanjing, China) via the QuantStudio 6 (Thermo Fisher, Waltham, MA, USA). The primers named as gene-QF/gene-QR were used to perform the experiment (Appendix A). The β2-tubulin coding gene was used as an endogenous control for normalization. All the reactions consisted of three biological replicates.

### 2.6. Appressorium Formation Assay

Polyester with a thickness of 25 μm was placed on water agar; then aliquots of the conidia suspension were inoculated onto the polyester to induce the appressorium formation. After incubation at 28 °C for 8 and 18 h, conidia germination behavior and appressorium formation were investigated under a microscope. At least 100 conidia were observed to calculate the appressorium formation rate. Each strain contained three biological replicates and the experiment was repeated twice.

### 2.7. Melanin Content Measurement

The melanin content was measured with the Fungal melanin quantification kit (GENMED SCIENTIFICS INC, Wilmington, DE, USA). Briefly, the conidia of the strains were inoculated into a liquid minimal medium supplemented with glutamine (10 mmol L^−1^) as nitrogen source with the initial concentration of 10^4^ conidia mL^−1^, and incubated at 160 rpm, 28 °C for 3 d. The mycelium was collected and grounded to a fine powder in liquid nitrogen. Then, 0.1 g of the powder was used for melanin extraction according to the manufacture’s protocol. After that, the melanin content was quantitated at 360 nm using a spectrophotometer (Eppendorf, Donau City, Germany). Both WT and the mutant contained three biological replicates.

### 2.8. Chromatin Immunoprecipitation (ChIP) and Quantitative PCR Analysis

ChIP analysis was performed according to our previous work with some modification [37]. The FLAG-tagged mutant strain was used for the ChIP assays. The protoplast was prepared and adjusted to the concentration of 10^7^ cell mL^−1^, with a total volume of 10 mL. Genomic DNA and protein were cross-linked with 0.75% of formaldehyde for 10 min and the reaction was then stopped using 125 mmol L^−1^ glycine. Then, the sample was added with ChIP lysis buffer, and the nuclei were isolated by centrifugation at 4 °C, 1000× *g* for 10 min. After re-suspended with ChIP shearing buffer, the chromosome was sonicated for 5 min, with the pulse of 8 s sonication and 5 s interval, to an average size of 200–500 bp via a Vibra-Cell Processors (Sonics, Newtown, CT, USA). A small aliquot of sonicated chromatin was used as the input DNA control. After that, the chromatin solution was immunoprecipitated using Anti-FLAG^®^ M2 Magnetic Beads (Sigma-Aldrich, Louis, MA, USA) for 12 h at 4 °C with rotation; the sample incubated with IgG2a Magnetic Beads (MBL BEIJING BIOTECH, Beijing, China) was used as a mock/negative control. After the reverse of the cross-link and DNA purification, the enrichment of the promoter regions surrounding heat stress elements (HSEs) were determined by real-time quantitative PCR. Each sample contained three replicates and the experiment was conducted twice. The primers used are listed in Appendix A.

### 2.9. Recombinant Protein Preparation and Electrophoretic Mobility Shift Assay (EMSA)

EMSA analysis was performed as described by Ream et al. [38]. The coding sequence of *Cghsf1* was amplified, digested with *Kpn*I and *Hind*III, and cloned into the same sites of pCOLD vector. The recombinant vector was transformed into *Escherichla coli* strain BL21 (DE3). The expression of the recombinant proteins was induced by isopropyl β-D-1-thiogalactopyranoside at 16 °C and purified with Ni-NTA Superflow (QIAGEN, Venlo, The Netherlands) according to the manufacturer’s instructions. Oligonucleotide probes containing the HSE element were synthesized based on sequences of the target gene promoters. A total of 40 ng of double-stranded probes and 10 mg of protein were mixed in 50 μL volume and incubated at 25 °C for 2 h. After incubation, the samples were loaded into 2% (*w/v*) agarose gel and electrophoresis was conducted using 0.5 × TB buffer at 10 V/cm. After staining with ethidium bromide (EB), the agarose gel was visualized using a GelDoc imager system (BIO-RAD, Hercules, CA, USA).

### 2.10. Statistical Analysis

Data with a single variable were analyzed by one-way ANOVA, and mean separations were performed by Duncan’s multiple range test. Differences at *p* < 0.05 were considered significant.

## 3. Results

### 3.1. Identification and Bioinformatic Analysis of CgHSF1

The HSF1 protein coding gene *Cghsf1* was identified in the *C. gloeosporioides* genomic sequence. *Cghsf1* contains a 2181 bp open reading frame separated by one intron, and it encodes the protein CgHSF1 of 727 amino acids (Appendix A). To investigate the similarity relationships of CgHSF1 between other fungal HSF1 proteins, an unrooted phylogenetic tree was constructed (Figure 1A). The phylogenetic analysis indicated that HSF1 proteins are highly conserved in plant pathogenic fungi, and CgHSF1 has a higher similarity to that of other *Colletotrichum* genus. Through searching against the Pfam database, CgHSF1 was found to contain a conserved HSF type DNA-binding domain at the 137–238 amino acids (Figure 1B).

### 3.2. Subcellular Localization of CgHSF1

To investigate the subcellular localization of CgHSF1, the CgHSF1-sGFP tagged strain was constructed (Figure 2A and Appendix A). The strain expressing sGFP, which is driven by the promoter of glyceraldehyde-3-phosphate dehydrogenase gene from *Aspergillus nidulans* (PgpdA), was used as control check (CK). A thin layer of yeast casein sucrose (YCS) medium (1g L^−1^ yeast extract, 1g L^−1^ acid hydrolyzed casein, 2% *w/v* sucrose, 0.5% *w/v* agar, pH 6.9) was plated into dimples of slides. Then, a conidia suspension in ddH_2_O was inoculated on the medium and the slides were kept in Petri dishes with lids at 28 °C for 12–16 h. After incubation, the slide was observed under a confocal. The results showed that the fluorescence of the CK was located in both the cell nucleus and cytoplasm, whereas that of CgHSF1-sGFP was mainly located in the cell nucleus, as evidenced by the co-localization of 4′,6-diamidino-2-phenylindole (DAPI) staining (Figure 2B).

### 3.3. Generation of the PniiA-Cghsf1 Strain

We firstly tried to construct the *Cghsf1* knock-out mutant via a recombination strategy (Appendix A), but the results of three round transformation attempts were unsuccessful; coupled with the previous reports [39], we concluded that *Cghsf1* is essential for the viability of *C. gloeosporioides*. Therefore, we generated the mutant strain in which *Cghsf1* was under the control of a repressive promoter PniiA (Figure 3A and Appendix A). Three independent PCR diagnoses and sequencings verified that the two recombinant fragments were correctly integrated into the locus between “ATG” of *Cghsf1* and its native promoter; and the expression cassette of PniiA-Cghsf1 was also correct in sequence (Figure 3B). After purification by a single conidia isolation, the mutant strain was named as PniiA-*Cghsf1*. After that, the transcription levels of *Cghsf1* in the PniiA-*Cghsf1* strain during in vitro and in planta stage were investigated. Through qRT-PCR, we found that the expression level of *Cghsf1* in PniiA-*Cghsf1* was about 4.27-fold lower than WT during in planta stage, whereas when cultured on medium in vitro, the transcripts of *Cghsf1* in the mutant were 4.21-fold higher than that of WT (Figure 3C). These results suggested that in the mutant strain PniiA-*Cghsf1*, *Cghsf1* was significantly repressed in planta.

### 3.4. CgHSF1 Is Required for Pathogenicity

To analyze the role of CgHSF1 in the pathogenicity of *C. gloeosporioides* to *Hevea*, the conidia of the strains were inoculated on detached leaves with or without wounds. The results showed that WT strain caused typical anthracnose symptoms on the intact leaves with a disease incidence of 55%, and the lesions were about 7 mm in diameter at 3 dpi; whereas the PniiA-*Cghsf1* caused no lesions on the intact leaves (Figure 4A–C). When inoculated onto the pre-wounded leaves, both WT and PniiA-*Cghsf1* successfully infected the leaves (Figure 4D,E); however, the lesion diameters caused by PniiA-*Cghsf1* were merely 2 mm, in comparison with that of 8 mm in WT (Figure 4F). These results suggested CgHSF1 is required for the pathogenicity of *C. gloeosporioides* through the regulation of both the initial infection process and the following hyphae development in host cells.

### 3.5. CgHSF1 Plays a Role in Appressorium Formation

To investigate whether CgHSF1 is involved in appressorium formation, the conidia of PniiA-*Cghsf1* was inoculated onto polyester and the germination behavior was analyzed. The results showed that WT and PniiA-*Cghsf1* had similar germination rates after incubation for 8 h (Figure 5B). In addition, about 73% of the WT conidia formed typical appressoria, together with the melanin deposition at the appressorium cell wall, whereas only 23% conidia of PniiA-*Cghsf1* formed appressoria with abnormal shapes and longer germ tubes; furthermore, there is little melanin deposition in these abnormal appressoria. After incubation for another 10 h, over 80% of the WT conidia formed appressoria with obvious melanism; in comparison, only 34% conidia of PniiA-*Cghsf1* formed appressoria, most of which are with abnormal shapes (Figure 5A,C).

### 3.6. CgHSF1 Plays a Role in Melanin Biosynthesis

To investigate whether CgHSF1 plays a role in melanin biosynthesis, the melanin biosynthesis in the PniiA-*Cghsf1* mutant was observed and measured. WT and PniiA-*Cghsf1* strains were cultured on solidified minimal medium supplemented with different nitrogen sources for 6 d. The results (Figure 6A) showed that when cultured on the rich medium with yeast extract as the nitrogen source, WT and the PniiA-*Cghsf1* mutant showed a higher growth rate than that cultured with the other nitrogen sources, and the colonies of the two strains were both with obvious melanism at 3 and 6 d. When cultured on a medium with NaNO_3_ as the sole nitrogen source, the mutant showed a decrease in colony growth rate than WT (Figure 6B); besides, both WT and the mutant showed obvious melanism at 6 d. When cultured on minimal medium with ammonium tartrate and glutamine, WT colonies showed a strong melanism at 3 d; in comparison, those of the PniiA-*Cghsf1* showed only a little melanism; in addition, after culture for another 3 d, the PniiA-*Cghsf1* mutant still produced obviously less melanin than WT on the medium with glutamine. Then, WT and the PniiA-*Cghsf1* mutant strains were incubated in a liquid medium supplemented with glutamine, and the melanin content was quantitated. After incubation in the liquid medium for 3 d, the mycelium of WT produced 485 μg melanin per g of fresh mycelia, while that of PniiA-*Cghsf1* was only 202 (Figure 6C). These results suggest that CgHSF1 plays a role in melanin biosynthesis.

### 3.7. CgHSF1 Regulates Transcription of Melanin Biosynthesis Genes

There was a total of eight proteins involved in melanin biosynthesis being identified in *C. gloeosporioides*, including the transcription factor CMR1, polyketide synthase (PKS), polyketide shortening (YG), HN reductase (T4HRa and T4HRb), scytalone dehydratase (SCD), multicopper oxidase (T3HR), and laccase (Appendix A). To investigate whether CgHSF1 regulates the transcription of these melanin biosynthesis pathway genes, the qRT-PCR was conducted. The results showed that these genes except for *T3HR* were significantly down-regulated in both the appressoria and in planta hyphae of PniiA-*Cghsf1* (Figure 7).

### 3.8. CgHSF1 Directly Binds to the Promoters of Melanin Biosynthesis Genes

The presence of HSE (GAAnTTC and GAAnnTTC) elements was examined in the 2 kb upstream regions of these melanin biosynthetic genes. The *in silicon* analysis showed that seven of eight genes except *PKS* contain at least one HSE element in their promoters. Therefore, we performed a ChIP-qPCR using the CgHSF1-FLAG tagged strain. The chromatin pellet was immunoprecipitated using Anti-FLAG, and the enrichment of promoter fragments containing HSEs was expressed as the percentage relative to the input DNA. The results show an enrichment in the promoter regions of four genes, *CMR1, YG, T4HRa,* and *SCD*, compared with when nonspecific antibodies (IgG2a) were used (Figure 8). Additionally, the control experiment with the promoter region of *β2-tubulin* showed that there is no non-specific enrichment for the sequences that do not contain HSE elements. Furthermore, the bindings of CgHSF1 to the corresponding probes containing HSEs of the four candidate genes were confirmed by the gel shift bands of the EMSA assay (Figure 9). Taken together with the gene expression analysis that revealed that the expression of melanin-related genes was lower in the mutant than in WT, these data suggest that CgHSF1 directly binds to the promoters and activates the transcription of these four melanin-biosynthesis-related genes.

## 4. Discussion

Pathogenic fungi have evolved sophisticated mechanism to infect their hosts by producing virulence factors, including effectors, enzymes, and secondary metabolites. The expression of the related genes is under tight regulation to assure successful infection and colonization. Our goal was to determine if the HSF transcription factor modulates virulence in *C. gloeosporioides*. The phylogenetic analysis and protein domain analysis revealed that CgHSF1 is highly conserved in phytopathogenic fungi, and it contains a typical HSF-type DNA-binding domain. The microscopic analysis showed the specific subcellular location of CgHSF1 in the nucleus. These results confirmed that CgHSF1 is a transcription factor. To investigate its biological functions, we firstly tried to construct the *Cghsf1* knock-out mutant. However, the failures of the attempts suggested that CgHSF1 is essential for the viability *C. gloeosporioides*. Similar results were also found in HSFs of *C. albicans*, which are required for core gene expression [39]. In many filamentous fungi, nitrite reductase (*niiA*) and nitrate reductase (*niaD*) genes were inducible by the presence of nitrate when cultured in vitro [40,41,42]. In this study, we found that the transcription of *CgniiA* was dramatically repressed when cultured with glutamine as the sole nitrogen source and during the in planta stage (Appendix A). Therefore, the promoter of *CgniiA* (PniiA) was used to modulate the expression of *Cghsf1*, and the mutant PniiA-*Cghsf1* was constructed by inserting the PniiA promoter between the coding sequence of *Cghsf1* and its native promoter. The following qRT-PCR analysis showed that the expression level of *Cghsf1* in PniiA-*Cghsf1* was induced by NaNO_3_ and repressed by glutamine when cultured in vitro; moreover, the expression level of *Cghsf1* was reduced about fourfold than that of WT during in planta colonization.

Then, the pathogenicity of PniiA-*Cghsf1* was assayed to investigate the role of CgHSF1 in the infection process. Here, the Hevea leaves were inoculated in two ways, with one group of leaves being pre-wounded and the other non-wounded. When inoculated onto the wounds of leaves, PniiA-*Cghsf1* could infect the leaves and formed typical anthracnose symptoms; however, the lesion diameter caused by the mutant was significantly smaller than that of WT. Microorganisms that colonize plants are continually challenged with plant defense responses, of which the oxidative burst, the rapid release of reactive oxygen species (ROS), is one of the earliest responses of plant hosts against pathogens at the invasion site [43,44]. In our previous work, rubber tree leaves were found to have a strong ROS burst after the infection of *C. gloeosporioides* [36]. Moreover, HSF1 and its downstream target HSP90 is well known to play a primary and global role in stress responses in eukaryotic organisms [19,45]. In the soybean stem and root rot pathogen Phytophthora sojae, PsHSF1 is critical for the pathogenicity of the pathogen by detoxifying the plant’s oxidative burst [46]. Thus, we speculated that the attenuated pathogenicity of PniiA-*Cghsf1* was mainly due to the decreased tolerance to oxidative stress. Meanwhile, when inoculated onto the intact leaves, PniiA-*Cghsf1* caused no lesion at all, suggesting that the mutant lost ability to penetrate the leaves. *Colletotrichum* species breach the intact cuticles of plant hosts via the formation of the specialized structures called appressoria [3]. So, the appressorium formation of PniiA-*Cghsf1* was surveyed. The results showed that depletion of *Cghsf1* significantly decreased appressorium formation; moreover, PniiA-*Cghsf1* only formed appressoria with abnormal shapes and relative long germ tubes, and there is little melanin deposition in these abnormal appressoria. A series of proteins are already known for being involved in appressorium formation in pathogenic fungi, including cell cycle control regulators [47], autophagy components [48], MAP kinase pathway and the cAMP response pathway [49,50,51]. In the human pathogen *C. albicans*, HSF1 is required for virulence via controlling cell transition from yeast to filamentous growth [34]. These results enlightened us that CgHSF1 might control the cell developmental transition from germ tube to appressorium via interaction with these pathways, although there are different mechanisms in pathogenicity between *C. albicans* and plant pathogenic fungi.

It is interesting that melanin deposition was significantly impaired in the abnormal appressoria of PniiA-*Cghsf1*. There are two well-characterized pathways for melanin biosynthesis, the DOPA pathway and the DHN pathway [52,53]; and most of plant pathogenic fungi synthesize melanin via the DHN pathway. Melanin is well known for its biological function in the detoxification of ROS and protection against environmental stresses in living cells. In plant pathogenic fungi, melanin plays a myriad of biological roles in morphogenesis, virulence, and energy transduction [54,55]. To investigate whether the depletion of *Cghsf1* impairs melanin biosynthesis, the PniiA-*Cghsf1* mutant was cultured on a solidified or in liquid minimal medium supplemented with different nitrogen sources. The colony growth analysis showed that induction or repression of transcription of *Cghsf1* did not impair the vegetative growth, but significantly influenced the melanism in the PniiA-*Cghsf1* mutant. When the expression of *Cghsf1* was induced by nitrate, the colony melanism was observational increased in the mutant compared with WT, while when *Cghsf1* was repressed by glutamine, the colony melanism was significantly reduced in the mutant, which was in accordance with the melanin quantitation results. Notably, the melanization of appressoria is indispensable for the successful infection of intact leaves, for it is required for development of high pressures within appressoria [56] and/or cell-wall rigidity in appressoria of the corn pathogen [57]. Then, the effect of CgHSF1 on the transcription of melanin biosynthesis enzymes was analyzed via a qRT-PCR analysis. The results showed that in the PniiA-*Cghsf1* mutant, transcripts of six of seven melanin biosynthesis genes were significantly reduced in the appressoria, and all seven genes were reduced in the in planta tissues.

In fungi, the genes involved in secondary metabolite biosynthesis are frequently physically linked in the genome, which are named as clusters; in addition, expression of these gene clusters depends on specialized TFs [58]. Here, we found that transcription of the TF CMR1, the transcriptional activators of melanin biosynthesis [59,60], was down-regulated in PniiA-*Cghsf1*. Although HSF proteins display a wide variability between different species, the heat shock element (HSE) that targeted by HSFs are well conserved. The HSE consists of three contiguous repeats of the short sequence 5′-nGAAn-3′ [61]. The examination of the promoter regions of these seven melanin biosynthesis genes and CMR1 showed that *CMR1*, *YG*, *T4HRa*, and *SCD*, contained at least one HSE element (Appendix A). The following ChIP-PCR and EMSA analysis confirmed that the four genes were direct targets of CgHSF1. Although the other four genes, *PKS*, *T4HRb*, *T3HR*, and *Laccase* were not identified as the direct targets of CgHSF1, their transcriptions were also under the regulation of CgHSF1. We speculated that it is because that these four genes are the direct targets of CMR1 [52,53], and CgHSF1 could regulate their transcription in an indirect way. These findings revealed that, in addition to CMR1, CgHSF1 may also function as transcriptional activators of melanin biosynthesis.

Taken together, we conclude that CgHSF1 is involved in the pathogenicity of *C. gloeosporioides* through the activation of melanin biosynthesis genes and the regulation of appressorium formation (Figure 10). Our study extends the understanding of fungal HSF1 proteins, especially their involvement in pathogenicity.

## Figures and Tables

**Figure 1 jof-08-00175-f001:**
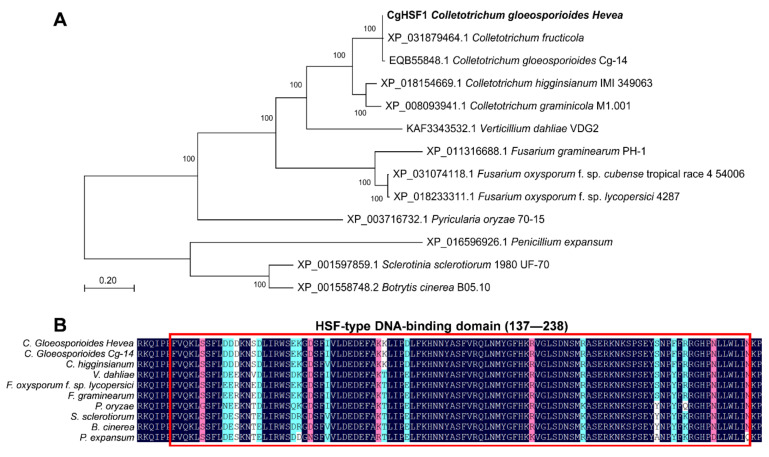
Bioinformatic analysis of CgHSF1. (**A**) The phylogenetic relationship of CgHSF1 and HSF1 proteins from some other plant pathogenic fungi. (**B**) Comparison of the conserved HSF DNA binding domain of fungal HSF1 proteins.

**Figure 2 jof-08-00175-f002:**
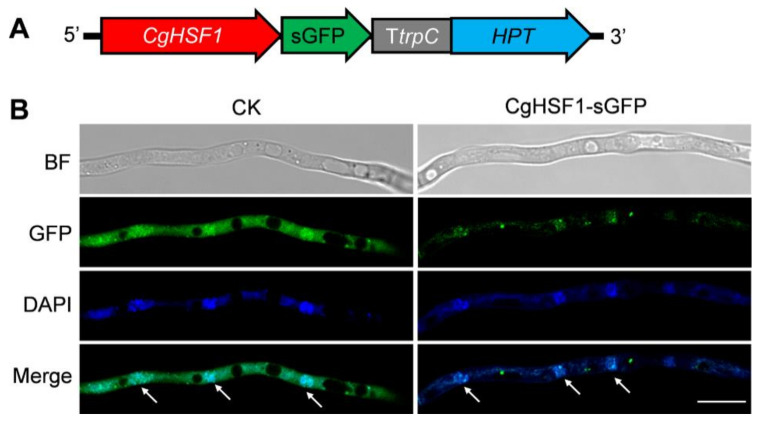
The subcellular localization of CgHSF1. (**A**) Expression cassette of CgHSF1-sGFP. (**B**) Fluorescence microscopes of hyphae. The arrowheads indicate nuclei. BF: bright field. DAPI: 4′,6-diamidino-2-phenylindole. Scale Bar = 25 μm.

**Figure 3 jof-08-00175-f003:**
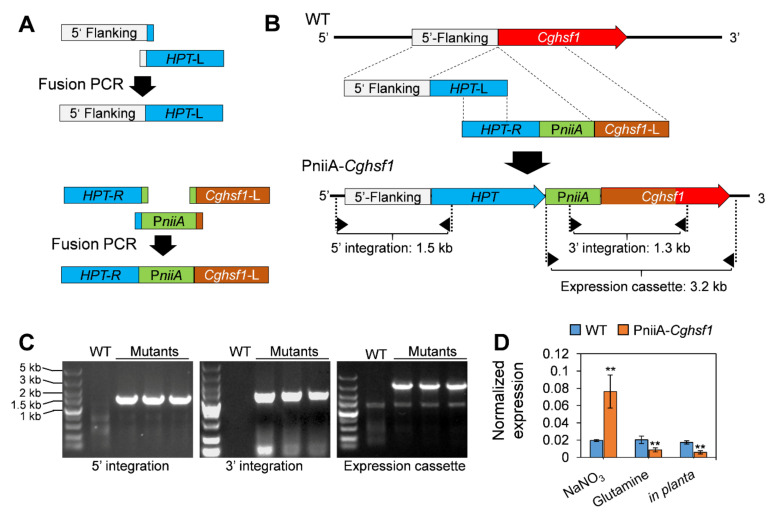
Construction of the repressive strain PniiA-*Cghsf1*. (**A**) Strategy for overlap PCR. The overlap nucleotides were indicated with the colors corresponding to relative fragments. *Cghsf1*-L indicate the nucleotides of 5′ part of *Cghsf1* with about 800 bp. (**B**) Strategy for the construction of PniiA-*Cghsf1* via recombination (**C**) Independent PCR diagnosis for the integration of the two recombinant fragments into the target locus, and diagnosis for correctness of the expression cassette. (**D**) Normalized expression level of *Cghsf1* in WT and the mutant in response to nitrogen sources and during in planta stage. The β2-tubulin coding gene was used as an endogenous control for normalization. Values are shown as the means ± standard deviations (SD). Asterisks indicate significant difference at *p* < 0.01 (**).

**Figure 4 jof-08-00175-f004:**
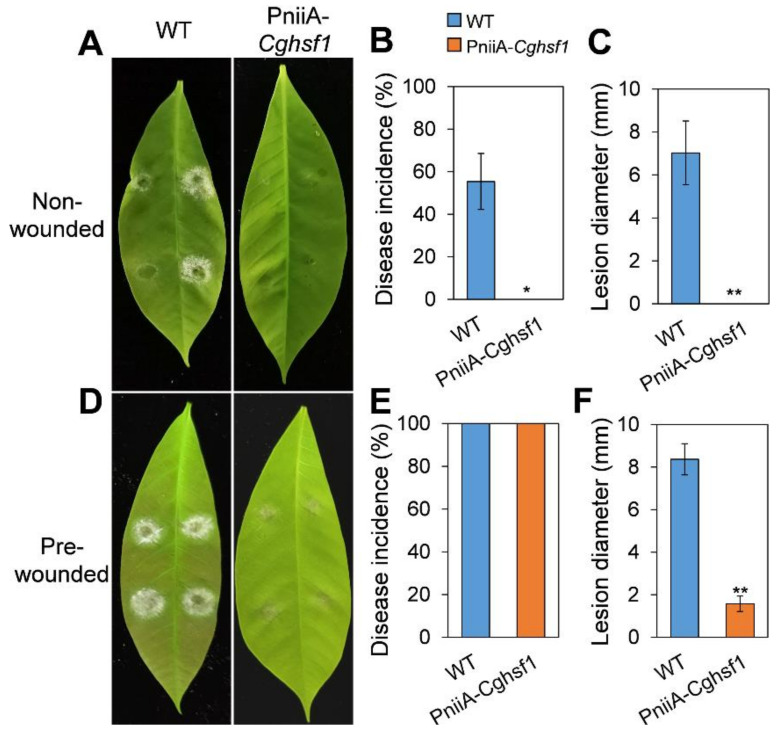
Pathogenicity assay of PniiA-*Cghsf1*. Figures and the data were captured at 3 d post inoculation. Disease symptoms of the intact rubber tree leaves (**A**) and pre-wounded leaves (**D**). Disease incidence of the intact rubber tree leaves (**B**) and pre-wounded leaves (**E**). Lesion diameters in the intact rubber tree leaves (**C**) and pre-wounded leaves (**F**). Values are shown as the means ± standard deviations (SD) of three groups of samples. Asterisks indicate significant difference at *p* < 0.05 (*) and *p* < 0.01 (**).

**Figure 5 jof-08-00175-f005:**
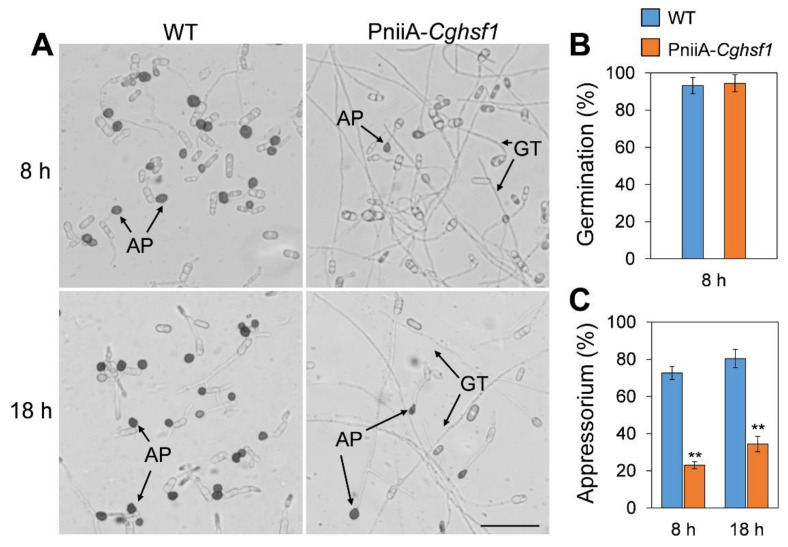
Conidia germination and appressorium formation analysis. (**A**) Microscopes of conidia after incubation on polyester for 8 and 18 h. AP: appressoria. GT: germ tube. Bar = 50 μM. (**B**) Conidia germination rates of WT and PniiA-*Cghsf1* strains. (**C**) Appressorium formation rates of WT and PniiA-*Cghsf1* strains. Values are shown as the means ± standard deviations (SD) of three groups of samples. Asterisks indicate significant difference at *p* < 0.01 (**).

**Figure 6 jof-08-00175-f006:**
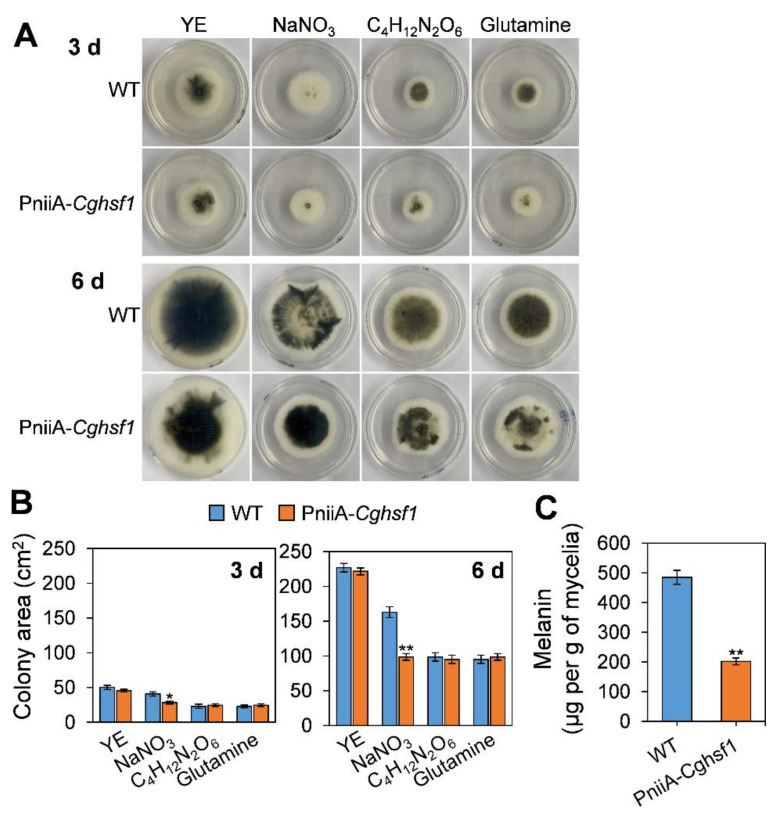
Growth analysis and measurement of melanin biosynthesis. (**A**) Colony morphology (bottom) of WT and the PniiA-*Cghsf1* mutant after growth on minimal medium supplemented with yeast extract (YE, 2 g L^−1^), NaNO_3_ (20 mmol L^−1^), ammonium tartrate (C_4_H_12_N_2_O_6_) (10 mmol L^−1^), and glutamine (10 mmol L^−1^) as nitrogen sources. (**B**) Colony area after growth on minimal medium for 3 and 6 d. (**C**) Quantitation of melanin after growth in a liquid minimal medium supplemented glutamine for 3 d. Columns with asterisks indicate significant difference at *p* < 0.05 (*) and *p* < 0.01 (**).

**Figure 7 jof-08-00175-f007:**
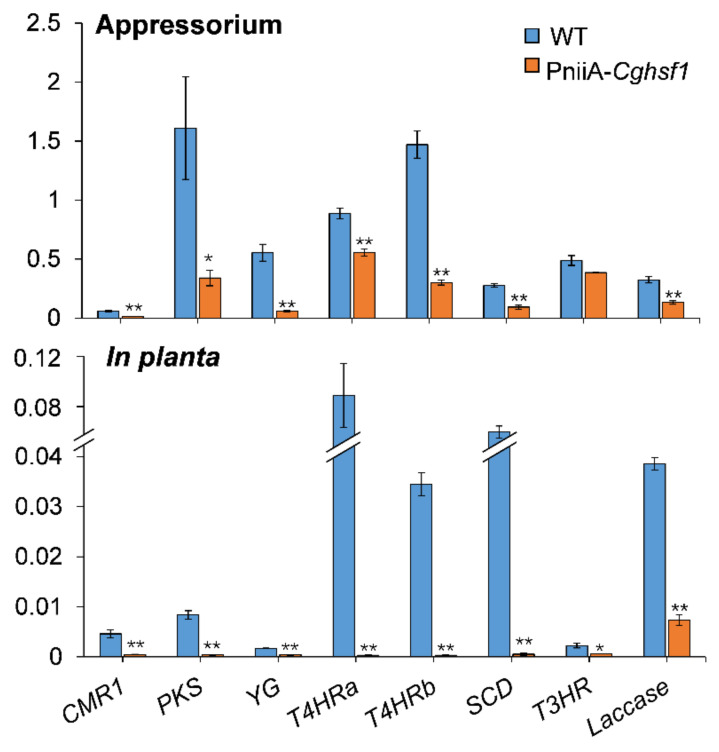
Normalized expression levels of melanin biosynthesis pathway genes in the appressoria and in planta stages of PniiA-*Cghsf1*. The β2-tubulin coding gene was used as an endogenous control for normalization. Values are shown as the means ± standard deviations (SD). Asterisks indicate significant difference at *p* < 0.05 (*) and *p* < 0.01 (**).

**Figure 8 jof-08-00175-f008:**
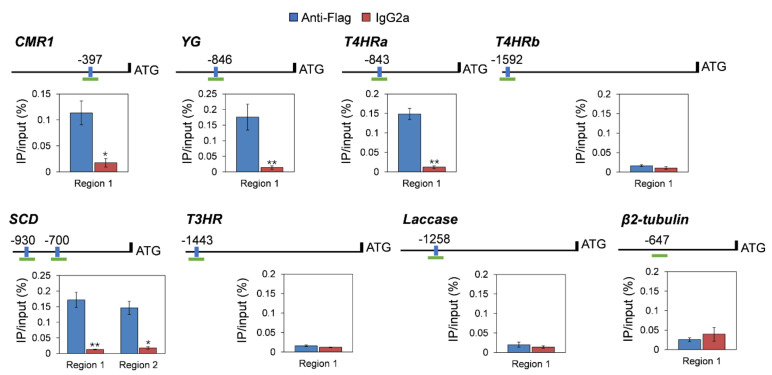
ChIP-qPCR assays indicated that CgHSF1 directly binds to the promoters of four melanin biosynthesis genes. The promoter structures are shown in the schematic diagrams. Blue boxes with numbers represent the positions of HSE elements to the translational start site. Green fragments represent the regions used for ChIP-qPCR. Values are the percentage of DNA fragments that were coimmunoprecipitated with specific anti-FLAG antibodies or non-specific antibodies relative to the input DNA. Error bars represent the standard deviations (SD) of three independent experiments. Asterisks indicate significant difference at *p* < 0.05 (*) and *p* < 0.01 (**) between samples co-immunoprecipitated with anti-FLAG and IgG2a.

**Figure 9 jof-08-00175-f009:**
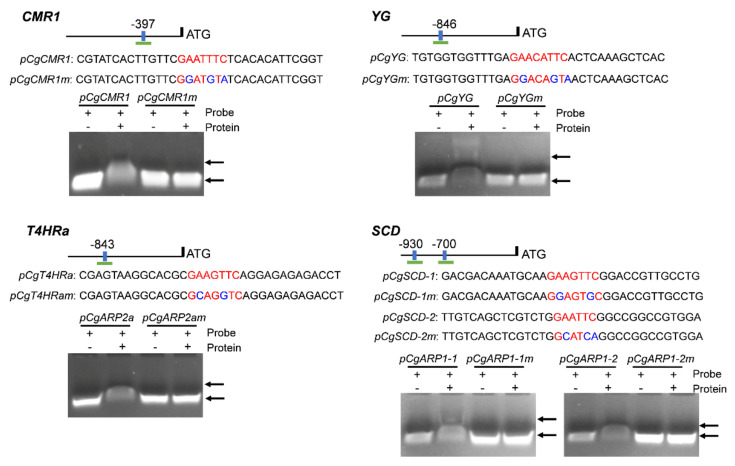
Gel mobility shift assays revealed that CgHSF1 directly binds to the HSE elements in the promoter region of four melanin biosynthesis genes. The native probe and mutated probe sequences corresponding to the promoters are shown, with red letters representing the HSE elements, and blue letters representing the mutated bases. Arrowheads indicate the specific complexes.

**Figure 10 jof-08-00175-f010:**
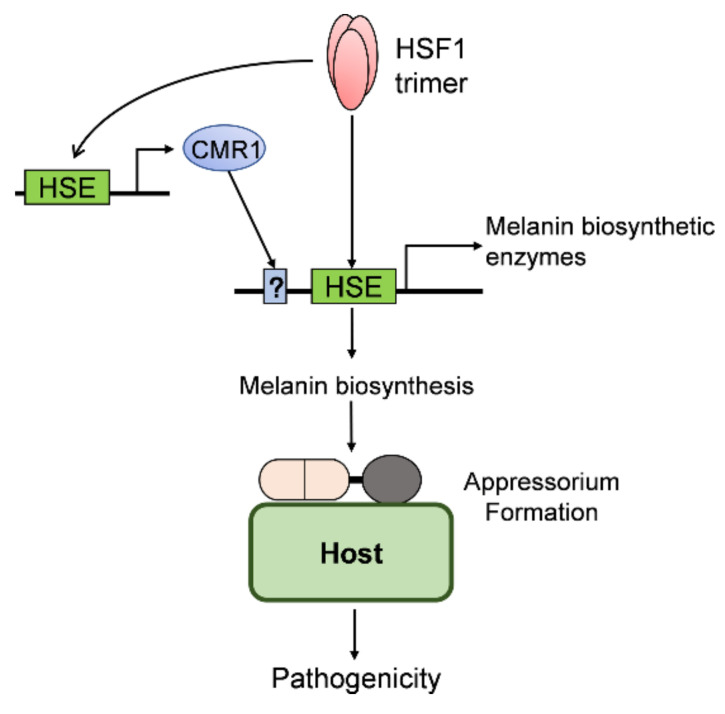
The proposed model of the CgHSF1 in the regulation of the pathogenicity of *C. gloeosporioides*. HSE: heat shock element.

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
