# Peer review of "Heat Shock Transcription Factor CgHSF1 Is Required for Melanin Biosynthesis, Appressorium Formation, and Pathogenicity in Colletotrichum gloeosporioides"

_jof, 2022, doi:10.3390/jof8020175_

Round 1

Reviewer 1 Report

This is a review of “Heat shock transcription factor CgHSF1 is required for melanin biosynthesis, appressorium formation, and pathogenicity in Colletotrichum gloeosporioides”.

This manuscript provides some insight in the role of a gene named CgHSF1 by the authors in pathogenicity and, appressorium formation and expression of genes involved in melanin synthesis. The way the research was conducted and the paper is written appear rather straightforward, although some details are missing about the reasoning or execution of some steps. In general, I would recommend a good look at the order in which information/methods are presented, sometimes the choice for a certain approach or used material only becomes evident in the supplementary material mentioned later in the manuscript.

  • The use of the promoter with altered expression (PniiA) is an ingenious approach to obtain knock-down mutants. I would consider calling it a repressible promoter since it is likely expressed during transformation/recovery and suppressed during the condition of interest (infection). Please discuss the information (either your conditions for qPCR or literature sources) about the expression of the specific CgniiA used in more detail.
    1. Make it clear whether PniiA was inserted between the endogenous promoter and the ATG of CgHSF1 (line 95) or replaced a part of the promoter (line 311) if the latter, how much was replaced?
  • I would prefer to see the qPCR data displayed as normalised values (relative to normalising gene) to give a better sense of expression levels in all strains and conditions used. If relative expression is used, provide a very good description of the condition used as reference setting and how the relative values are obtained.
  • Can you compare the expression of genes during in vitro growth in the same medium as used for inoculation, besides or instead of the minimal medium mentioned.
    1. Also discus if the two media differentially affect of PniiA (and maybe even PCgHSF1)
    2. Bring the expression pattern of PniiA forward
  • Define minimal medium
  • Have a good look at the English, I can perfectly well understand most parts but there are a few points were the presence/absence of articles (the/a/an) or singular/multiple formulation might be confusing for some readers.
  • Describe how you identified all Cg genes mentioned in this manuscript, especially CgHSF1 and niiA, provide a database entry and/or locus Identifier (with the exact version of genome used for this study).
  • The CHiP analysis by qPCR of selected elements appears very biased, all elements you amplified are enriched. With this limited set of results it appears there always is enrichment in the FLAG-IP and there is no specificity. Presented like that it looks like you just worked to get the desired results. To demonstrate specificity you need to show that any other region is NOT enriched and the HSE elements you identified behave different from the rest of the genome. The best approach would be to sequence the whole pool of FLAG-IPed and IgG2A-IPed DNA to show which promoter regions are selectively expressed, without prior bias. You need to at least include some qPCR of genomic regions that are NOT expected to be enriched after FLAG IP to show specificity, such as the promoters and coding regions that are further away from the selected elements than the fragmentation size (line 156), promoters from the other melanin genes, any other gene, eg your housekeeping gene used for expression qPCR ( or the promoter thereof)
  • Line 112, clarify “two round PCR”, nested or two independent primer sets?
  • Line 131 describe the medium and temperature regime of the in vitro conditions, especially relevant of there are changes between culturing and inoculation.
  • Line 146 humidity (in petridishes with lid?) and medium used for conidia suspension.
  • Line 152 give a reference for the CHiP approach used.
  • Line 153 provide the total volume/cells used for isolation of nuclei
  • Include in Table S2 where primers are relative to HSE position.
  • Line 166 Define EMSA
  • Line 181 provide reference for genomic sequence and GeneID
  • Line 193-203 provide description of conditions for GFP analysis, medium, temperature, age germtube or appressoria?, define CK
  • Line 212 “and sequencing” > “by sequencing”
  • Figure 3 and S1, S2, S3 the dotted lines cross and thus insinuate that the initial PCR and overlap PCR used to make constructs invert some of the sequences, while from the text and resulting arrows this does not appear to be the case. Use parallel lines to indicate the exact limits of PCR amplification and junctions in overlap PCR.
    1. It might be an option to use crossed and differently dotted/interrupted lines to indicate recombination events between construct and genomic DNA in the generation of transformants.
    2. Define fragments in panels of Fig 3B
    3. It might be helpful to number all primers in Table S2 and insert these numbers in all these diagrams to make it easier for readers to follow the construction and verification.
    4. Define CgHSF1-L and CgHSF1-R in red arrows, in legend or in body text of Methods.
  • Figure 4 include graphs for disease incidence and severity (% spreading and size) for each inoculation method to support remarks in the text.
  • Figure 5 / line 242-257 Please discuss if the germination efficiency of the WT and mutant differ or are identical. Is the medium used inducing or suppressing PniiA?
  • Figure 6 define relative or show normalised data, which would also provide information on difference between AP and invivo expression in the WT.
  • Line 327 You speculated about effect of PniiA-Cghsf1 on pathogenicity via HSP90 and oxidative stress. You can easily turn this speculation into using your CHIP DNA in enrichment qPCR of the promoter of the gene you just mentioned……
  • Line 341 “there is different mechanism” > “there are different mechanisms”
  • Line 358 define YFs
  • line 371 its > their
  • Figure S4 give invitro medium and conditions. Again preferred normalised expression rather than relative, describe how relative was calculated.

Author Response

This manuscript provides some insight in the role of a gene named CgHSF1 by the authors in pathogenicity and, appressorium formation and expression of genes involved in melanin synthesis. The way the research was conducted and the paper is written appear rather straightforward, although some details are missing about the reasoning or execution of some steps. In general, I would recommend a good look at the order in which information/methods are presented, sometimes the choice for a certain approach or used material only becomes evident in the supplementary material mentioned later in the manuscript.

  1. The use of the promoter with altered expression (PniiA) is an ingenious approach to obtain knock-down mutants. I would consider calling it a repressible promoter since it is likely expressed during transformation/recovery and suppressed during the condition of interest (infection). Please discuss the information (either your conditions for qPCR or literature sources) about the expression of the specific CgniiA used in more detail.
    1. Make it clear whether PniiA was inserted between the endogenous promoter and the ATG of CgHSF1 (line 95) or replaced a part of the promoter (line 311) if the latter, how much was replaced?

Answer: Thanks for the suggestion, description of the promoter and the mutant strain was changed to “repressible promoter” and “repressible” mutant.

       In some other fungi such as Aspergillus fumigatus, the PniiA promoter was induced by nitrate while repressed by ammonium (Lamoth et al., 2012). In the previous study, to test whether CgniiA was induced by nitrate or repressed by ammonium, C. gloeosporioides WT strain was cultured on the minimal medium supplemented with NaNO3, Ammonium tartrate (C4H12N2O6), and Yeast extract as nitrogen source, respectively. Then the expression levels of CgniiA and Cghsf1 were analyzed. The results showed that, unlike that in Aspergillus fumigatus, the expression levels of CgniiA was not influenced significantly by different nitrogen source, and Ammonium tartrate showed no repression effect compared with NaNO3. Cghsf1 transcription was not influenced by these nitrogen source either.

       In addition, in our previous study, the transcriptome of C. gloeosporioides at in vitro (cultured on Czapek–Dox Medium) and in vivo stages were sequenced and analyzed (data not shown in the present manuscript). According to the results, the expression levels of Cghsf1 were 33 and 40 (FPKM) during in vitro and in vivo, respectively; while that of CgniiA were 154 and 2 respectively; and that of β2-tubulin were 740 and 1093 respectively. The results of RT-qPCR assay were in accordance with the transcriptomics data.

       Taken together, the expression of CgniiA and Cghsf1 are not influenced by these nitrogens source. Moreover, CgniiA transcription is significantly repressed during in vivo stage, suggesting that its promoter PniiA could be used for gene manipulation. The qRT-PCR results were added into the revised supplementary Figure S4, and the discussion of the information about the expression of CgniiA was added into the manuscript.

Answer to question a: In the present study, we wanted to insert the PniiA sequence right into the locus between the start codon “ATG” of Cghsf1 and its native promoter. So, the nucleotides right before the ATG and the 5’ part of Cghsf1 sequence were used as recombinant fragments (Figure 3A). As shown in revised Figure 3C, to confirm the integrations of the recombinant fragments to the right locus, two independent the PCR diagnosis were conducted using primer pairs surrounding the recombinant fragments. The PCR products were gel electrophoresis analyzed and sequenced for correctness. The results were in accordance with the expectation, suggesting that PniiA was inserted into the locus between native promoter and “ATG” of Cghsf1.

  1. I would prefer to see the qPCR data displayed as normalised values (relative to normalising gene) to give a better sense of expression levels in all strains and conditions used. If relative expression is used, provide a very good description of the condition used as reference setting and how the relative values are obtained.

Answer: According to your suggestion, All the qTR-PCR results were displayed as normalized values.

  1. Can you compare the expression of genes duringin vitro growth in the same medium as used for inoculation, besides or instead of the minimal medium mentioned?
    1. Also discus if the two media differentially affect of PniiA (and maybe even PCgHSF1)
    2. Bring the expression pattern of PniiA forward

Answer: As mentioned in Answer 1, the expression of CgniiA and Cghsf1 are not influence by nitrogen source, and the CgniiA transcription is repressed in vivo. Here malt extract broth at very low concentration was used to re-suspended the conidia. The suspension was used to inoculate the leaves since this approach could ensure the germination and the following infection of the conidia.

  1. Define minimal medium

Answer: Formula of minimal medium was added into the revised manuscript.

  1. Have a good look at the English, I can perfectly well understand most parts but there are a few points were the presence/absence of articles (the/a/an) or singular/multiple formulation might be confusing for some readers.

Answer: The manuscript was carefully reviewed and many errors were revised.

  1. Describe how you identified all Cg genes mentioned in this manuscript, especially CgHSF1 and niiA, provide a database entry and/or locus Identifier (with the exact version of genome used for this study).

Answer: The genome sequence of Colletotrichum gloeosporioides from Hevea was deposited in the NCBI database; however, the gene annotation data has not yet been uploaded and the accession number of CgHSF1 was missing here. To provide the sufficient information, the nucleotide and amino acid sequences were added into the Supplementary Table S1. And we will conduct the gene annotation as soon as possible.

  1. The CHiP analysis by qPCR of selected elements appears very biased, all elements you amplified are enriched. With this limited set of results it appears there always is enrichment in the FLAG-IP and there is no specificity. Presented like that it looks like you just worked to get the desired results. To demonstrate specificity you need to show that any other region is NOT enriched and the HSE elements you identified behave different from the rest of the genome. The best approach would be to sequence the whole pool of FLAG-IPed and IgG2A-IPed DNA to show which promoter regions are selectively expressed, without prior bias. You need to at least include some qPCR of genomic regions that are NOT expected to be enriched after FLAG IP to show specificity, such as the promoters and coding regions that are further away from the selected elements than the fragmentation size (line 156), promoters from the other melanin genes, any other gene, eg your housekeeping gene used for expression qPCR ( or the promoter thereof)

Answer: Thanks for the suggestion. Actually, in the present study, the 2.0 kb upstream sequences of the genes were selected as promoters and screened for the presence of HSE elements. The results showed that T4HRb, T3HR, and Laccase have candidate HSE elements at -1592, -1443, and -1258 bp locus in the promoters. However, the following ChIP assay revealed that there is no enrichment of promoter fragments of the three genes. Thus, we speculated that these three HSE elements were not involved in the transcriptional regulation and omitted the results.

       To provide sufficient information, the ChIP results of these three genes were added into the revised Figure 7, and the description was also included into the text.

  1. Line 112, clarify “two round PCR”, nested or two independent primer sets?

Answer: The description was revised to “two independent PCR diagnosis”.

  1. Line 131 describe the medium and temperature regime of the in vitroconditions, especially relevant of there are changes between culturing and inoculation.

Answer: All the strains were kept on PDA at 28°C. For protoplast transformation, culture in minimal medium, and incubation after inoculation, the strains were also kept at 28°C. Description of the temperature regime was added into the manuscript.

  1. Line 146 humidity (in petridishes with lid?) and medium used for conidia suspension.

Answer: The mistake has been revised to “petri dishes with lid”. Conidia suspension was prepared by re-suspended in 0.5% malt extract broth.

  1. Line 152 give a reference for the CHiP approach used.

Answer: ChIP analysis was performed according to our previous work with some modification. And the reference was added into the manuscript.

  1. Line 153 provide the total volume/cells used for isolation of nuclei

Answer: A total volume of 10 mL protoplasts were used for the following treatment and nucleus isolation. The information was added.

  1. Include in Table S2 where primers are relative to HSE position.

Answer: The positions of the primers were added into Table S2.

  1. Line 166 Define EMSA

Answer: The definition of EMSA was added.

  1. Line 181 provide reference for genomic sequence and GeneID

Answer: As mentioned above, the nucleotide and amino acid sequences were added into the Supplementary to make the information clearer; and we will conduct the gene annotation as soon as possible

  1. Line 193-203 provide description of conditions for GFP analysis, medium, temperature, age, germtube or appressoria?, define CK

Answer: A thin layer of Yeast Casein Sucrose (YCS) medium (1g L-1 yeast extract, 1g L-1 acid hydrolyzed casein, 2% w/v sucrose, 0.5% agar, pH 6.9) was plated into dimples of slides. The conidia suspension in ddH2O was inoculated on the medium and the slides were kept in petri dishes with lid at 28°C for 12-16 h. Then the slide was observed under confocal. Description of the method was added into the manuscript.

       Definition of “Control check (CK)” was added.

  1. Line 212 “and sequencing” > “by sequencing”

Answer: The mistake was revised.

  1. Figure 3 and S1, S2, S3 the dotted lines cross and thus insinuate that the initial PCR and overlap PCR used to make constructs invert some of the sequences, while from the text and resulting arrows this does not appear to be the case. Use parallel lines to indicate the exact limits of PCR amplification and junctions in overlap
  2. It might be an option to use crossed and differently dotted/interrupted lines to indicate recombination events between construct and genomic DNA in the generation of transformants.
  3. Define fragments in panels of Fig 3B
  4. It might be helpful to number all primers in Table S2 and insert these numbers in all these diagrams to make it easier for readers to follow the construction and verification.
  5. Define CgHSF1-L and CgHSF1-R in red arrows, in legend or in body text of Methods.

Answer:

       The overlap PCR strategy was illustrated in the revised Figure 3A.

       The expected PCR fragment sizes were marked in Figure 3B, and the results of gel electrophoresis were revised.

       The primer names were added into the method part, and application of the primers were annotated with more details in Table S2

       Cghsf1-L and Cghsf1-R were illustrated in brown color in the revised Figure 3 and S2, S3. And the description of the fragments was added into the Method part and Figure legends.

  1. Figure 4 include graphs for disease incidence and severity (% spreading and size) for each inoculation method to support remarks in the text

Answer: The data was added into the revised Figure 4.

  1. Figure 5 / line 242-257 Please discuss if the germination efficiency of the WT and mutant differ or are identical. Is the medium used inducing or suppressing PniiA?

Answer: The germination efficiency of the mutant showed no significant difference with that of WT. The data was added into the revised Figure 5.

For the experiment, conidia suspension in ddH2O was inoculated onto Polyester that placed on water agar. According to our qRT-PCR results, PniiA was repressed under this condition.

  1. Figure 6 define relative or show normalised data, which would also provide information on difference between AP and in vivo expression in the WT.

Answer: According to your suggestion, All the qTR-PCR results were displayed as normalized values.

  1. Line 327 You speculated about effect of PniiA-Cghsf1 on pathogenicity via HSP90 and oxidative stress. You can easily turn this speculation into using your CHIP DNA in enrichment qPCR of the promoter of the gene you just mentioned……

Answer: It is widely accepted that HSF1 proteins could regulate expression of HSPs; in addition, we mainly focus on the function of CgHSF1 in the regulation of melanin biosynthesis in the present study. Therefore, we did not conduct the relative assay.

  1. Line 341 “there is different mechanism” > “there are different mechanisms”

Answer: The mistake was revised.

  1. Line 358 define YFs

Answer: It should be “TFs” here. The mistake was revised.

  1. line 371 its > their

Answer: The mistake was revised.

  1. Figure S4 give in vitro medium and conditions. Again preferred normalised expression rather than relative, describe how relative was calculated.

Answer: As mentioned above, the in vitro medium and conditions were added into the manuscript, and the qRT-PCR results were displayed as normalised expression.

Reviewer 2 Report

Comments

In the study by Gao et al. a heat shock transcription factor was functionally characterized, the authors indicate that CgHSF1 is essential for the survival of Colletotrichum gloeosporioides. Furthermore, the inducible mutant shows a significant decrease in pathogenicity, defects in melanin biosynthesis, and appressorium formation; however considerations must be addressed to improve the work. Below listed my comment and suggestion.

  1. A supplementary figure with the sequences in nucleotides and amino acids of the CgHSF1 must be incorporated.
  2. The accession number for CgHSF1 should be added in materials and methods and in figure 1B.
  3. Improve figure 1A. indicate the sequence length for DNA-binding domain.
  4.  I will suggest the authors perform an alignment (supplementary figure), highlighting the DNA binding domains for the proteins used in the phylogeny (figure 1B).
  5. Improve the description of the methods, for example:
    • Line 91 the enzyme used to amplify the DNA is not indicated.
    • Line 97 describe the PCR fusion method.
    • Line 132 the DNA polymerase used for Quantitative RT-PCR analysis is not indicated.
    • Line 155 the process for isolating the nuclei is not indicated.
    • Line 156 indicate sonication conditions.
    • Line 167 improve the description of cloning since the conditions of protein induction and purification are not indicated.

Author Response

In the study by Gao et al. a heat shock transcription factor was functionally characterized, the authors indicate that CgHSF1 is essential for the survival of Colletotrichum gloeosporioides. Furthermore, the inducible mutant shows a significant decrease in pathogenicity, defects in melanin biosynthesis, and appressorium formation; however considerations must be addressed to improve the work. Below listed my comment and suggestion.

  1. A supplementary figure with the sequences in nucleotides and amino acids of the CgHSF1 must be incorporated.

Answer: According to your suggestion, the sequences were added into the Table S1 in the Supplementary.

  1. The accession number for CgHSF1 should be added in materials and methods and in figure 1B.

Answer: The genome sequence of Colletotrichum gloeosporioides from Hevea was deposited in the NCBI database; however, the gene annotation data has not yet been uploaded. Therefore, the accession number of CgHSF1 was missing here. The nucleotide and amino acid sequences were added into the supplementary; and we will conduct the gene annotation as soon as possible.

  1. Improve figure 1A. indicate the sequence length for DNA-binding domain. I will suggest the authors perform an alignment (supplementary figure), highlighting the DNA binding domains for the proteins used in the phylogeny (figure 1B)

Answer: Thanks for the suggestions. We re-performed an alignment, re-ordered the Figure 1, and highlighting the DNA binding domains of the selected proteins in the revised Figure 1B.

  1. Improve the description of the methods, for example:
    • Line 91 the enzyme used to amplify the DNA is not indicated.

Answer: The information of the enzyme (TransStart® FastPfu DNA Polymerase (TransGen Biotech, China)) was added into the manuscript.

  • Line 97 describe the PCR fusion method.

Answer: A brief description of the PCR fusion was added: “Firstly, the upstream flanking fragments of Cghsf1 was amplified with primer pairs HSF-5F/HSFhpt-5MR , with the reverse primer has 17 nucleotides (nts) complementary to HPT sequence; and a truncated HPT fragment was amplified with primer pairs HSFhpt-5MF/hpt-SPLR, with the forward primer has 17 nts complementary to Cghsf1; then the two fragments were was ligated together via fusion PCR with the two fragments as templates. Secondly, the other truncated HPT (with primers hpt-SPLF/hptniiA-MR), the PniiA promoter (with hptniiA-MF/niiAHSF-MR), and the 800 bp of 5′ part of Cghsf1 sequence (with niiAHSF-MF/HSF-L-R) were amplified, and then the three fragments were fused with PCR.”

  • Line 132 the DNA polymerase used for Quantitative RT-PCR analysis is not indicated.

Answer: The ChamQ SYBR Color qPCR Master Mix (Vazyme, China) was used to perform the experiment, and the information was added into the manuscript.

  • Line 155 the process for isolating the nuclei is not indicated.

Answer: The method was added: “Then sample was added with ChIP lysis buffer, and the nuclei were isolated by cen-trifugation at 4°C, 1000 g for 10 min.”

  • Line 156 indicate sonication conditions.

Answer: The methods for isolating the nuclei and the sonication were added into the manuscript: “Then sample was added with ChIP lysis buffer, and the nuclei were isolated by centrifugation at 4°C, 1000 g for 10 min. After re-suspended with ChIP shearing buffer, the chromosome was sonicated for 5 min, with the pulse of 8 s sonication and 5 s interval, to an average size of 200–500 bp via a Vibra-Cell Processors (Sonics, USA).”

  • Line 167 improve the description of cloning since the conditions of protein induction and purification are not indicated.

Answer: The brief methods for protein induction and purification were added into the manuscript: “The coding sequence of Cghsf1 was amplified, digested with KpnI and HindIII, and cloned into the same sites of pCOLD vector. The recombinant vector was transformed into Escherichla coli strain BL21 (DE3). Expression of the recombinant proteins was induced by isopropyl β-D-1-thiogalactopyranoside at 16°C and purified with Ni-NTA Superflow (QIAGEN, USA) according to the manufacturer’s instructions.”

Reviewer 3 Report

The manuscript entitled “Heat shock transcription factor CgHSF1 is required for melanin biosynthesis, appresorium formation, and pathogenicity in Colletotrichum gloeosporioides” by Gao et al analyzes the role of CgHSF1 in pathogenicity and tries to shed light on the mechanisms underpinning that role.

For this purpose, the authors first identified the Cghsf1 gene in the C. gloeosporioides  genome, and confirmed that the encoded transcripton factor shows a nuclear localization by means of fluorescence microscopy. Afterwards, the authors tried to obtain the null mutant following a gene replacement strategy based on the use of the SUR gene from M. oryzae, which confers resistance to the herbicide chlorimuron ethyl. However, the authors state that after several transformation experiments no mutants could be obtained. It is not clear whether the failure was caused by the lack of activity of the resistance gene or other causes. The conclusion of the authors is that Cghsf1 is essential for viability. This conclusion lacks of experimental support.

The second attempt consisted on the obtention of a mutant with a replacement of the native Cghsf1 allele. The allele would have been replaced by a cassette consisting in the flanking region of the Cghsf1 gene followed by the coding region of the hygromycin resistance gene (HPT), and the native Cghsf1 gene under the control of the inducible promoter of the nitrite reductase gene (PniiA). Proof of the correct identity of the transformants obtained is showed in Fig. 3. This experiment raises serious doubts about the nature of the putative mutants. First, the PCRs showing the amplicons corresponding to the flanking-HPRT region and the PniiA-Cghsf1 region are not conclusive about the correct sizes of these two fragments. There is a lack of correct markers in the agarose gel showed in Panel B of Fig. 3. The amplicons could have any size between 1 and 2 Kbs. Second, and more important, the amplification of those two fragments do not demonstrate the replacement of the native allele of Cghsf1 by the inducible mutant allele. In order to show this the authors need to perform aditional PCRs using pairs of primers targeting the flanking regions of the integration and a Southern hybridization that confirms the loss of the native allele and the presence of the inducible one. Aditionally, the strategy followed in this second attempt raises the question on why the hygromycin resistance gene was not used to obtain the knock out in the first attempt. The authors plead that full length of Cghsf1 in the inducible strain was sequenced and verified (in order to show that the allele is fully functional, I guess). However, the full cassette sequence is needed and not only the Cghsf1 allele.

Panel C in Fig. 3 shows the results of RT-qPCR aimed to demonstrate the differences of expression between the WT Cghsf1 allele and the inducible one, both in vitro and in planta (not in vivo, as the authors state). These results also raise serious doubts:

  1. Why the WT allele shows a higher expression during in vitro growth than during the infection of the hevea leaves? I would expect exactly the opposite result should the encoded trancription factor behave as a pathogenicity factor.
  2. What is the inducer of the PniiA promoter, present in the growing medium and absent in the pathogenicty experiment, which is responsible for the increased expression of Cghsf1 during in vitro growth? It is assumed by the authors (as stated in the Discussion, but not in the Results section) that this inducer is nitrate, which seems logic. However, this asumption lacks of experimental support. The authors should compare the expression of the inducible allele in medium with different concentrations of nitrate as sole nitrogen source, in order to validate their assumption.

The authors do not show any evidence on possible morphological and functional changes induced in C. gloeosporioides  by a reduction of expression of Cghsf1. An impairment of the growth capability of the mutant would also explain the results obtained in the pathogenicity assays. A phenotypic characterization of some of the mutants obtained is required.

The rest of the experiments reported in the present work deal with the presumptive role of CgHSF1 in appresorium formation and the regulation of the melanin biosynthesis genes. These experiments are not conclusive if the true genetic nature of the inducible mutants is not clearly demonstrated.

Other issues of concern:

  1. Although a complete table of the primers used is included as Supplementary Material, the specific primers used for the obtention of amplicons in sections 2.3, 2.5 and 2.7 should be provided in the text to facilitate the understanding of the experiments and figures.
  2. It is recommended a through revision of the writing as some sentences are meaningless or difficult to understand. Examples:
  • L. 96-97: “Firstly, the upstream flanking fragments of Cghsf1 was ligated with the truncated HPT using fusion PCR”. How many fragments were ligated?
  • L. 115-117: “After that, the heterokaryon of the correct transformants were purified by single conidia isolations”. How many nuclei contain the C. gloeosporioides conidia?. The mutants used in the study are heterokaryons composed of nuclei with the native allele of CgHSF1 and nuclei with the inducible allele?
  • L. 185-186. “To investigate the evolutionary link of CgHSF1 between other fungal HSF1 proteins, an unrooted phylogenetic tree was constructed”. Unrooted trees do not give any information on evolutionary relationships, but on similarity relationships.

In conclusion, the central issue of this work is a very interesting question: what is the role of the transcription factor CgHSF1 in the pathogenicity mechanisms of C. gloeosporioides towards hevea? However, the work suffers from several experimental drawbacks that should be corrected before publication. Mainly, the absence of experimental verification of the correct integration of the inducible allele in the genome of the fungus and the lack of an adequate phenotypic characterization of the putative mutants.

Author Response

The manuscript entitled “Heat shock transcription factor CgHSF1 is required for melanin biosynthesis, appresorium formation, and pathogenicity in Colletotrichum gloeosporioides” by Gao et al analyzes the role of CgHSF1 in pathogenicity and tries to shed light on the mechanisms underpinning that role.

  1. For this purpose, the authors first identified the Cghsf1 gene in the C. gloeosporioides  genome, and confirmed that the encoded transcripton factor shows a nuclear localization by means of fluorescence microscopy. Afterwards, the authors tried to obtain the null mutant following a gene replacement strategy based on the use of the SUR gene from M. oryzae, which confers resistance to the herbicide chlorimuron ethyl. However, the authors state that after several transformation experiments no mutants could be obtained. It is not clear whether the failure was caused by the lack of activity of the resistance gene or other causes. The conclusion of the authors is that Cghsf1 is essential for viability. This conclusion lacks of experimental support.

Answer: The acetolactate synthase gene (SUR, also named as ILV2SUR) was an effective selective marker for gene knock-out in several fungi, such as Magnaporthe oryzae and Botrytis cinerea (Sweigard et al., 1997; Yang et al., 2013; Yang and Naqvi, 2014). And SUR was more effective than hygromycin resistance gene (HPT) in selection for transformants of C. gloeosporioides. In the protoplast transformation, herbicide chlorimuron ethyl at 100 ug mL-1 could effectively select transformants containing SUR gene from wild type strain with relatively low false positive rate; whereas for HPT, hygromycin at concentration up to 300 ug mL-1 was needed for transformants selection. In addition, according to our previous work, SUR was proved effective in gene knock-out of C. gloeosporioides (Wang et al., 2018).

       In addition, unlike that in animal or plants, fungi usually contain only one HSF coding gene. And as the core transcription factors, HSFs were found essential for viability of several fungi, for example C. albicans (Sorger and Pelham, 1988; Nicholls et al., 2009; ). Besides, HSF1 proteins are also required for viability of mice and Drosophila (Xiao et al., 1999; Guertin and Lis, 2010).

       Based on these reasons, we concluded that the failure of getting the knock-out mutant is because that CgHSF1 is essential for viability of C. gloeosporioides.

  • Sweigard, J., Chumley, F., Carroll, A., Farrall, L., and Valent, B. (1997) A series of vectors for fungal transformation. Fungal Genet Newsl 44: 52–53.
  • Yang F, Naqvi NI. Sulfonylurea resistance reconstitution as a novel strategy for ILV2-specific integration in Magnaporthe oryzae. Fungal Genet Biol. 2014
  • Yang Q, Jiang J, Mayr C, Hahn M, Ma Z. Involvement of two type 2C protein phosphatases BcPtc1 and BcPtc3 in the regulation of multiple stress tolerance and virulence of Botrytis cinerea. Environ Microbiol. 2013
  • Nicholls, S.; Leach, M.D.; Priest, C.L.; Brown, A.J. Role of the heat shock transcription factor, Hsf1, in a major fungal pathogen that is obligately associated with warm-blooded animals. Microbiol. 2009, 74, 844–861.
  • Wang, Q.; An, B.; Hou, X.; Guo, Y.; Luo, H.; He, C. Dicer-like proteins regulate the growth, conidiation, and pathogenicity of Colletotrichum gloeosporioides from Hevea brasiliensis. Microbiol. 2018, 8, 2621.
  • Sorger, P.K. and Pelham, H.R. (1988) Yeast heat shock factor is an essential DNA-binding protein that exhibits temperature-dependent phosphorylation. Cell, 54, 855–864
  • Xiao X, Zuo X, Davis AA, McMillan DR, Curry BB, Richardson JA, Benjamin IJ. HSF1 is required for extra-embryonic development, postnatal growth and protection during inflammatory responses in mice. EMBO J. 1999 Nov 1;18(21):5943-52.
  • Guertin MJ, Lis JT. Chromatin landscape dictates HSF binding to target DNA elements. PLoS Genet. 2010 Sep 9;6(9):e1001114.

  1. The second attempt consisted on the obtention of a mutant with a replacement of the native Cghsf1 allele. The allele would have been replaced by a cassette consisting in the flanking region of the Cghsf1 gene followed by the coding region of the hygromycin resistance gene (HPT), and the native Cghsf1 gene under the control of the inducible promoter of the nitrite reductase gene (PniiA). Proof of the correct identity of the transformants obtained is showed in Fig. 3. This experiment raises serious doubts about the nature of the putative mutants. First, the PCRs showing the amplicons corresponding to the flanking-HPRT region and the PniiA-Cghsf1 region are not conclusive about the correct sizes of these two fragments. There is a lack of correct markers in the agarose gel showed in Panel B of Fig. 3. The amplicons could have any size between 1 and 2 Kbs. Second, and more important, the amplification of those two fragments do not demonstrate the replacement of the native allele of Cghsf1 by the inducible mutant allele. In order to show this the authors need to perform aditional PCRs using pairs of primers targeting the flanking regions of the integration and a Southern hybridization that confirms the loss of the native allele and the presence of the inducible one. Aditionally, the strategy followed in this second attempt raises the question on why the hygromycin resistance gene was not used to obtain the knock out in the first attempt. The authors plead that full length of Cghsf1 in the inducible strain was sequenced and verified (in order to show that the allele is fully functional, I guess). However, the full cassette sequence is needed and not only the Cghsf1 allele.

Answer: In the present study, we wanted to insert the PniiA sequence right into the locus between the start codon “ATG” of Cghsf1 and its native promoter. So, the nucleotides right before the ATG and the 5’ part of Cghsf1 sequence were used as recombinant fragments (Figure 3A). As shown in revised Figure 3C, to confirm the integrations of the recombinant fragments to the right locus, two independent the PCR diagnosis were conducted using primer pairs surrounding the recombinant fragments.        Besides, in the last version of manuscript, the DNA marker in the DNA gel was not clear, so the PCR diagnosis was re-conducted using the three independent mutants. The results showed that the diagnosis fragments were in accordance with expectations, with 5 diagnosis bands of 1.5 kb, and 3’ of 1.3 kb (revised Figure 3C).

       In addtion, according to the suggestion, additional PCR (revised Figure 3B) was conducted to confirm the correctness of the expression cassette, with one primer complementary to the beginning of PniiA and the other primer complementary to down-stream sequence of Cghsf1. The gel electrophoresis showed the DNA fragment of the mutants were 3.2 kb, and no fragments were amplified in the WT (despite of the false positive bands). Moreover, the nucleotide sequencing showed that the fragment was in accordance with the expectation. These results suggested that PniiA was inserted into the locus between native promoter and “ATG” of Cghsf1.

       For the question that why SUR was selected for gene knock-out and HPT for gene knock repression, as mentioned in Answer to question 1, SUR is more effective than HPT for transformants selection. However, SUR sequence is with the length of 2.8 kb while HPT is only 1.4 kb; therefore, to ensure the nucleotide correctness and the insert of the fragments into the expected locus between the promoter and start codon “ATG” of Cghsf1, HPT was selected for the promoter insertion manipulation.

  1. Panel C in Fig. 3 shows the results of RT-qPCR aimed to demonstrate the differences of expression between the WT Cghsf1 allele and the inducible one, both in vitro and in planta (not in vivo, as the authors state). These results also raise serious doubts:

Why the WT allele shows a higher expression during in vitro growth than during the infection of the hevea leaves? I would expect exactly the opposite result should the encoded trancription factor behave as a pathogenicity factor.

What is the inducer of the PniiA promoter, present in the growing medium and absent in the pathogenicty experiment, which is responsible for the increased expression of Cghsf1 during in vitro growth? It is assumed by the authors (as stated in the Discussion, but not in the Results section) that this inducer is nitrate, which seems logic. However, this asumption lacks of experimental support. The authors should compare the expression of the inducible allele in medium with different concentrations of nitrate as sole nitrogen source, in order to validate their assumption.

Answer: In our previous study, the transcriptome of C. gloeosporioides at in vitro (cultured on Czapek–Dox Medium) and in vivo stages were sequenced and analyzed (data not shown in the present manuscript). According to the results, the expression level of Cghsf1 was 33 and 40 (FPKM) during in vitro and in vivo, respectively; while that of CgniiA were 154 and 2 respectively; and that of β2-tubulin were 740 and 1093 respectively.

       The result showed that CgniiA was with higher transcription level than Cghsf1 during in vitro, and with lower expression than Cghsf1 during in vivo. These results were in accordance with the RT-qPCR assay. In addition, the activities of HSFs are usually regulated at the protein level through phosphorylation and formation of trimer but not at transcription level (Sorger, and Pelham, 1988).

       For the inducer of the PniiA, in some other fungi such as Aspergillus fumigatus, the PniiA promoter was induced by nitrate while repressed by ammonium (Lamoth et al., 2012). In the previous study, to test whether CgniiA was induced by nitrate or repressed by ammonium, C. gloeosporioides WT strain was cultured on the minimal medium supplemented with NaNO3, Ammonium tartrate (C4H12N2O6), and Yeast extract as nitrogen source, respectively. Then the expression levels of CgniiA was analyzed. The results showed that, unlike that in Aspergillus fumigatus, the expression levels of CgniiA was not influenced significantly by different nitrogen source, and Ammonium tartrate showed no repression effect compared with NaNO3. The results were added into the revised supplementary Figure S4.

       Taken together, the expression of CgniiA was significantly repressed during in vivo stage.

  • Lamoth F, Juvvadi PR, Fortwendel JR, Steinbach WJ. Heat shock protein 90 is required for conidiation and cell wall integrity in Aspergillus fumigatus. Eukaryot Cell. 2012 Nov;11(11):1324-32.
  • Sorger, P.K. and Pelham, H.R. (1988) Yeast heat shock factor is an essential DNA-binding protein that exhibits temperature-dependent phosphorylation. Cell, 54, 855–864

  1. The authors do not show any evidence on possible morphological and functional changes induced in C. gloeosporioides by a reduction of expression of Cghsf1. An impairment of the growth capability of the mutant would also explain the results obtained in the pathogenicity assays. A phenotypic characterization of some of the mutants obtained is required.

The rest of the experiments reported in the present work deal with the presumptive role of CgHSF1 in appresorium formation and the regulation of the melanin biosynthesis genes. These experiments are not conclusive if the true genetic nature of the inducible mutants is not clearly demonstrated.

Answer: The experiment was firstly designed to investigated whether repression of Cghsf1 could interfere the vegetative growth or conidiation of C. gloeosporioides, however, as mentioned above, the PniiA promoter is only repressed significantly during in vivo in C. gloeosporioides. Therefore, only phenotypes of PniiA-Cghsf1 involved in pathogenicity processes were analyzed.

Other issues of concern:

  1. Although a complete table of the primers used is included as Supplementary Material, the specific primers used for the obtention of amplicons in sections 2.3, 2.5 and 2.7 should be provided in the text to facilitate the understanding of the experiments and figures.

Answer: According to the suggestion, the primer names were added into the method part to illustrate the PCR procedure. Moreover, the strategy for overlap PCR was added into the revised Figure 3A.

  1. It is recommended a through revision of the writing as some sentences are meaningless or difficult to understand. Examples:
  2. 96-97: “Firstly, the upstream flanking fragments of Cghsf1 was ligated with the truncated HPT using fusion PCR”. How many fragments were ligated?

Answer: The the strategy for overlap PCR was added into the revised Figure 3A; and description of the fusion PCR were revised for better understanding. “Firstly, the upstream flanking fragments of Cghsf1 was amplified with primer pairs HSF-5F/HSFhpt-5MR , with the reverse primer has 17 nucleotides (nts) complementary to HPT sequence; and a truncated HPT fragment was amplified with primer pairs HSFhpt-5MF/hpt-SPLR, with the forward primer has 17 nts complementary to Cghsf1; then the two fragments were was ligated together via fusion PCR with the two fragments as templates. Secondly, the other truncated HPT (with primers hpt-SPLF/hptniiA-MR), the PniiA promoter (with hptniiA-MF/niiAHSF-MR), and the 800 bp of 5′ part of Cghsf1 sequence (with niiAHSF-MF/HSF-L-R) were amplified, and then the three fragments were fused with PCR.”

  1. 115-117: “After that, the heterokaryon of the correct transformants were purified by single conidia isolations”. How many nuclei contain the C. gloeosporioides conidia? The mutants used in the study are heterokaryons composed of nuclei with the native allele of CgHSF1 and nuclei with the inducible allele?

Answer: After the protoplast transformation and diagnosis, the correct mutants are usually heterokaryon which contain both WT and mutant nuclei in C. gloeosporioides. And like most filamentous fungi, C. gloeosporioides conidia was found to contain only one nucleus in our previous study, therefore, the single conidia isolation was necessary to get the homozygote mutant.

  1. 185-186. “To investigate the evolutionary link of CgHSF1 between other fungal HSF1 proteins, an unrooted phylogenetic tree was constructed”. Unrooted trees do not give any information on evolutionary relationships, but on similarity relationships.

Answer: The expression was revised to “similarity relationships”. Besides, the manuscript was revised carefully all-through and some other errors were revised.

In conclusion, the central issue of this work is a very interesting question: what is the role of the transcription factor CgHSF1 in the pathogenicity mechanisms of C. gloeosporioides towards hevea? However, the work suffers from several experimental drawbacks that should be corrected before publication. Mainly, the absence of experimental verification of the correct integration of the inducible allele in the genome of the fungus and the lack of an adequate phenotypic characterization of the putative mutants.

Round 2

Reviewer 1 Report

This is a review for “Heat shock transcription factor CgHSF1 is required for melanin 2 biosynthesis, appressorium formation, and pathogenicity in 3 Colletotrichum gloeosporioides”, jof-1504614 v2.

 The revised version is a rapid and good improvement on the first version I have seen, most procedures have been clarified, either by re-ordering, extended explanation, inserted references or added data. Some of the revisions may have to be double checked for their formulation, e.g. how they sit in the context of the sentence they were made in….

I am glad all experiments were performed at the same temperature.

I do appreciate the data and discussion on gene-expression in response to nitrogen and in planta of both CgHSF1 and CGniia. I suppose the minimal medium for line 145 and line 150) are the same. Is there a carbon source?

The modification of most figures is sufficiently clear to me.

The addition of more panels in Figure 7, ChIP-qPCR is welcome, and interesting but still not addressing the point I made before. It might be a matter of scaling obfuscating the differences but the statistical analysis shown (**) insinuates that there still is differential enrichment of the putative around the (maybe imperfect) HSEs in T3HR, T4HRb and Laccase compared to IgG2a. The latter observations is actually in agreement with the expression data, so you could discuss that as well, for example on the impact of HSE variants on expression/binding. My major point is that you have not shown that for NON-HSE regions the ratio is identical. It might be assumed that the ratio is very close but you need to show it is actually the case in your material. You could choose primerpairs anywhere in the genome but if you have validated primers that are further away from the HSEregions than 2x the average fragment size it would be fine to use them, or your tubulin primers, or primers from your previous publications that are not HSF regulated.

Actually EMSA with the imperfect HSE would also be interesting, and to state that 4 genes are confirmed by EMSA (line407) it almost looks like you did the test for the other genes but don’t show the data.

Author Response

This is a review for “Heat shock transcription factor CgHSF1 is required for melanin 2 biosynthesis, appressorium formation, and pathogenicity in 3 Colletotrichum gloeosporioides”, jof-1504614 v2.

 The revised version is a rapid and good improvement on the first version I have seen, most procedures have been clarified, either by re-ordering, extended explanation, inserted references or added data. Some of the revisions may have to be double checked for their formulation, e.g. how they sit in the context of the sentence they were made in….

I am glad all experiments were performed at the same temperature.

  1. I do appreciate the data and discussion on gene-expression in response to nitrogen and in plantaof both CgHSF1 and CGniia. I suppose the minimal medium for line 145 and line 150) are the same. Is there a carbon source?

Answer: The missing data was added. Here 20 g L-1 sucrose was used as a carbon source. In addition, a colony growth assay was conducted by including glutamine as a sole nitrogen source according to the Reviewer 3’s suggestion. And the results showed that glutamine could significantly repress the expression of CgniiA and CgniaD; and thus, repressed the transcription of Cghsf1 in the mutant. The results were added into the revised Figure 6.

  1. The modification of most figures is sufficiently clear to me. The addition of more panels in Figure 7, ChIP-qPCR is welcome, and interesting but still not addressing the point I made before. It might be a matter of scaling obfuscating the differences but the statistical analysis shown (**) insinuates that there still is differential enrichment of the putative around the (maybe imperfect) HSEs in T3HR, T4HRb and Laccase compared to IgG2a. The latter observations is actually in agreement with the expression data, so you could discuss that as well, for example on the impact of HSE variants on expression/binding. My major point is that you have not shown that for NON-HSE regions the ratio is identical. It might be assumed that the ratio is very close but you need to show it is actually the case in your material. You could choose primer pairs anywhere in the genome but if you have validated primers that are further away from the HSE regions than 2x the average fragment size it would be fine to use them, or your tubulin primers, or primers from your previous publications that are not HSF regulated.

Answer: According to your suggestion, the promoter of β2-tubulin (the reference gene for qRT-PCR) was also selected the ChIP-qPCR analysis, and the result was added into the revised Figure 8. As expected, the result showed that there is no enrichment of the β2-tubulin promoter fragment in the immunoprecipitated chromatin pellet.

  1. Actually EMSA with the imperfect HSE would also be interesting, and to state that 4 genes are confirmed by EMSA (line407) it almost looks like you did the test for the other genes but don’t show the data.

Answer: It is a conventional method to analyze the target genes identified in the ChIP-qPCR with EMSA. Therefore, we did not synthesize the native and mutated probes of the other genes, and did not conduct EMSA test for the other genes.

Reviewer 3 Report

In the light of the answers given by the authors to some of the criticisms in my first review, the consideration of the work must be changed. The authors now inform that:

“In our previous study, the transcriptome of C. gloeosporioides at in vitro (cultured on Czapek–Dox Medium) and in vivo stages were sequenced and analyzed (data not shown in the present manuscript). According to the results, the expression level of Cghsf1 was 33 and 40 (FPKM) during in vitro and in vivo, respectively; while that of CgniiA were 154 and 2 respectively; and that of β2-tubulin were 740 and 1093 respectively.”

This is very important information that should have been included in the first version of the manuscript. Now we know that CgniiA is repressed during plant infection or, alternatively, that it is not induced, at least to the levels seen when the fungus is grown in culture (by the way, the authors should change the term “in vivo” for in “in planta” as they refer to what is happening during the plant infection process). The problem that now arises is why this differential expression. Theoretically, the gene coding for the nitrite reductase should be induced when nitrate is available as the sole nitrogen source, and repressed when there are preferred nitrogen sources (such as ammonium or glutamine). The data presented in Suplementary Figure 4 show similar expression rates for CgniiA, no matter the fungus is growing in nitrate, ammonium or yeast extract. The authors conclude in the Discussion that “the transcription of CgniiA was not influenced by nitrogen sources nitrate, ammonium, or yeast extract; instead the gene was dramatically reduced during in vivo in comparison with in vitro”. It is an obvious conclusion, but tells nothing about the repressor in planta (or the inducer in culture) which is a crucial question if the promoter has to be used for the purpose described in the work. I think that this aparent paradox is a problem derived of the very low expression levels of this gene in fungi. Nitrite is toxic, thus the conversion of nitrate to nitrite, previous to the reduction to ammonium, must be performed in such a way that the intracellular level of nitrite is always very low. Therefore, high expression levels of the fungal nitrite reductase coding genes are not needed. Besides, regulation of nitrogen asimilation is very complex in fungi, and it is subjected to several levels of regulation. In order to see induction of the genes involved in the nitrate asimilation pathway, glutamine is a better inducer than ammonium (see Pfanmüller et al, Frontiers in Microbiology,2017). I recommend to do again the experiment in Supp. Fig. S4 including glutamine as a sole nitrogen source and using the experimental design described in Pfanmüller et al.  Also, special care must be taken with the RT-qPCR analysis as very low expression rates are expected.

It is very important to determine how CgniiA is induced in culture because that would allow to experimentally modulate the expression of Cghsf1 when hooked up to the promoter of CgniiA. If the hypothesis of the authors is true, and Cghsf1 is esential for the survival of C. gloeosporoides, complete abolishion of the expression of Cghsf1 would determine the inability of the fungus to grow. Obviously, if the fungus is unable to grow it will be also unable to infect the host plant, but that does not make Cghsf1 a pathogenicty factor. The phenotypic characterization of the mutant when grown in culture would also shed light in some paradoxes shown in the work. If PniiA-Cghsf1 is unduced in culture and repressed in planta (the authors’ hypothesis), why the differences in appresorium formation between the WT and the mutant shown in Fig. 5? This experiment was performed with conidia germinated in water agar, and the most logic conclusion is that the inducer of CgniiA is absent from that medium.

Therefore it is important that the authors show the relationship between level of expression of PniiA-Cghsf1 and growth, and demonstrate that under the same level of expression seen in planta, the fungus may grow in culture. Alternatively, the authors might carry out a confocal microscopy study of the fungal growth during the host plant infection. In this way, the authors would be able to conclude that Cghsf1 has a role as a pathogenicty or virulence factor in C. gloeosporoides, and likely also demonstrate that is vital for the survival of this fungus.

Author Response

In the light of the answers given by the authors to some of the criticisms in my first review, the consideration of the work must be changed. The authors now inform that:

“In our previous study, the transcriptome of C. gloeosporioides at in vitro (cultured on Czapek–Dox Medium) and in vivo stages were sequenced and analyzed (data not shown in the present manuscript). According to the results, the expression level of Cghsf1 was 33 and 40 (FPKM) during in vitro and in vivo, respectively; while that of CgniiA were 154 and 2 respectively; and that of β2-tubulin were 740 and 1093 respectively.”

This is very important information that should have been included in the first version of the manuscript. Now we know that CgniiA is repressed during plant infection or, alternatively, that it is not induced, at least to the levels seen when the fungus is grown in culture (by the way, the authors should change the term “in vivo” for in “in planta” as they refer to what is happening during the plant infection process). The problem that now arises is why this differential expression. Theoretically, the gene coding for the nitrite reductase should be induced when nitrate is available as the sole nitrogen source, and repressed when there are preferred nitrogen sources (such as ammonium or glutamine). The data presented in Suplementary Figure 4 show similar expression rates for CgniiA, no matter the fungus is growing in nitrate, ammonium or yeast extract. The authors conclude in the Discussion that “the transcription of CgniiA was not influenced by nitrogen sources nitrate, ammonium, or yeast extract; instead the gene was dramatically reduced during in vivo in comparison with in vitro”. It is an obvious conclusion, but tells nothing about the repressor in planta (or the inducer in culture) which is a crucial question if the promoter has to be used for the purpose described in the work. I think that this aparent paradox is a problem derived of the very low expression levels of this gene in fungi. Nitrite is toxic, thus the conversion of nitrate to nitrite, previous to the reduction to ammonium, must be performed in such a way that the intracellular level of nitrite is always very low. Therefore, high expression levels of the fungal nitrite reductase coding genes are not needed. Besides, regulation of nitrogen asimilation is very complex in fungi, and it is subjected to several levels of regulation. In order to see induction of the genes involved in the nitrate asimilation pathway, glutamine is a better inducer than ammonium (see Pfanmüller et al, Frontiers in Microbiology,2017). I recommend to do again the experiment in Supp. Fig. S4 including glutamine as a sole nitrogen source and using the experimental design described in Pfanmüller et al.  Also, special care must be taken with the RT-qPCR analysis as very low expression rates are expected.

It is very important to determine how CgniiA is induced in culture because that would allow to experimentally modulate the expression of Cghsf1 when hooked up to the promoter of CgniiA. If the hypothesis of the authors is true, and Cghsf1 is esential for the survival of C. gloeosporoides, complete abolishion of the expression of Cghsf1 would determine the inability of the fungus to grow. Obviously, if the fungus is unable to grow it will be also unable to infect the host plant, but that does not make Cghsf1 a pathogenicty factor. The phenotypic characterization of the mutant when grown in culture would also shed light in some paradoxes shown in the work. If PniiA-Cghsf1 is unduced in culture and repressed in planta (the authors’ hypothesis), why the differences in appresorium formation between the WT and the mutant shown in Fig. 5? This experiment was performed with conidia germinated in water agar, and the most logic conclusion is that the inducer of CgniiA is absent from that medium.

Therefore it is important that the authors show the relationship between level of expression of PniiA-Cghsf1 and growth, and demonstrate that under the same level of expression seen in planta, the fungus may grow in culture. Alternatively, the authors might carry out a confocal microscopy study of the fungal growth during the host plant infection. In this way, the authors would be able to conclude that Cghsf1 has a role as a pathogenicty or virulence factor in C. gloeosporoides, and likely also demonstrate that is vital for the survival of this fungus.

Answer: Thanks for the suggestion.

According to your suggestion, we re-conducted the qRT-PCR by including glutamine as a sole nitrogen source. In addition, the preparation of the samples was also modified by shorting the incubation time to 1 d (Line 153); and the expression level analysis of nitrate reductase (niaD) was also included.

The qRT-PCR results showed that the transcription of niiA and niaD were both induced by NaNO3, and repressed by ammonium tartrate (about 2-fold) or glutamine (about 10-fold). Notably, expressions of niiA and niaD were also significantly repressed during in planta stage (Figure S4).

Meanwhile, we also re-analyzed the expression levels of Cghsf1. In WT strain, the expression levels of Cghsf1 showed no response to the nitrogen sources (Figure S4). While when cultured with NaNO3 as a sole nitrogen source, Cghsf1 in the PniiA-Cghsf1 mutant was with higher expression level than that in WT. Moreover, when cultured with glutamine as a sole nitrogen source or during in planta stage, Cghsf1 transcription in the PniiA-Cghsf1 mutant was significantly repressed (revised Figure 3). The expression pattern of Cghsf1 in the PniiA-Cghsf1 mutant was similar to that of niiA. The results suggested that the PniiA-Cghsf1 mutant could be used for the function analysis of Cghsf1.

In addition, we also re-conducted the growth assay (Figure 6).

When cultured on the rich medium with yeast extract as nitrogen source, C. gloeosporioides strains showed the highest growth rates; and the colonies of WT and the mutant showed similar morphology and growth rate, with obvious melanism at 3 d post inoculation.

When cultured on medium with ammonium tartrate or glutamine, the colony growth rate of C. gloeosporioides strains was lower compared with that on the yeast extract or NaNO3 as a sole nitrogen source. In addition, WT colonies were with a strong melanism; in comparison, that of the PniiA-Cghsf1 showed only a little melanism. To quantitate the melanin content, the strains were incubated in liquid minimal medium supplemented with glutamine for 3 d, and the mycelium were collected for the measurement. In accordance with the colony growth results, the melanin biosynthesis in the PniiA-Cghsf1 mutant was significantly reduced when the expression of Cghsf1 was repressed by glutamine.

Taken together, these results proved that CgHSF1 plays a role in melanin biosynthesis, and thus regulates pathogenicity of the pathogen. But how the activity of CgHSF1 protein was regulated remains unclear. According to the previous reports, protein phosphorylation and formation of trimer (as shown in Figure 10) are required for the activity of HSF1 proteins. So, we speculated that the phosphorylation of CgHSF1 is vital for its function in the regulation of melanin biosynthesis; but this would be another story.

Round 3

Reviewer 1 Report

This is a review for “Heat shock transcription factor CgHSF1 is required for melanin 2 biosynthesis, appressorium formation, and pathogenicity in 3 Colletotrichum gloeosporioides”, jof-1504614 v3.

The revision has again improved the manuscript, there are only several minor issues about careful description, formulation and discussion to be completed

  • Please re-evaluate the exact placement and orientation of dotted lines in the figures about strategy of building expression/replacement constructs.
    1. To me, crossing lines insinuates that the fragment is inserted in reverse orientation. Make it unambiguously clear in what orientation the fragments are stitched together.
    2. In Figure 3B it looks like the PniiA promoter fragment is INSERTED between the original promoter and the CDS of Cghsf1, is this correct? In your text you wrote replacement, which would imply that you removed part of the 5’-flanking region of Cghsf1. If there really is replacement you should describe how much was replaced.
  • The section on melanin quantification pulls together several of the observations made and can improve the coherence of the manuscript.
    1. Carefully revise the formulation of the melanin results for clarity (remove redundant observations) and English
    2. The statistics about the melanin quantification should be improved, provide information on the number of independent experiments etc.
    3. I would like to see a more complete integration of melanin quantification with te expression data, eg mention that higher expression of PniiA-hsf1 goes together with increased melanisation on nitrate medium
  • The model of HSF1-CMR1-HSE and final function is also helpful but should be presented as being only part of all regulation going.
    1. For reference, it is much easier to interpret supp Table 2 if you include a line with the canonical HSE sequence in the same table so people can see how similar or different each candidate HSE is
    2. Mention your interpretation of IP/input (%), how should readers read the graphs in fig8, and what are the implications of a certain anti-FLG vs IgG2A ratio. Explain how the tubulin control supports the interpretation of the data as specific binding at 5 other regions.
    3. Discuss more clearly that binding of hsf1 to HSE is not a pre-requisite for differential gene-expression, eg no binding in T4HRb, T3HR, Laccase (FIG8) but still differential expression for T4HRb and Laccase (FIG7). I know you kind of implied that in other fungi regulation of melanin genes goes through CMR1 but could explain that option in Cg more clearly in the text.

Author Response

The revision has again improved the manuscript, there are only several minor issues about careful description, formulation and discussion to be completed.

  • Please re-evaluate the exact placement and orientation of dotted lines in the figures about strategy of building expression/replacement constructs.
  1. To me, crossing lines insinuates that the fragment is inserted in reverse orientation. Make it unambiguously clear in what orientation the fragments are stitched together.

Answer: According to your suggestion, the crossing lines were changed to straight lines (Figure 3 and Figure S1-S3) to make the expression more excise.

  1. In Figure 3B it looks like the PniiA promoter fragment is INSERTED between the original promoter and the CDS of Cghsf1, is this correct? In your text you wrote replacement, which would imply that you removed part of the 5’-flanking region of Cghsf1. If there really is replacement you should describe how much was replaced.

Answer: The PniiA promoter is INSERTED between the original promoter and the CDS of Cghsf1. And the expression was revised to make the expression more excise (Line 98-99, 391-394).

  • The section on melanin quantification pulls together several of the observations made and can improve the coherence of the manuscript.
  1. Carefully revise the formulation of the melanin results for clarity (remove redundant observations) and English

Answer: The Figure 6B was removed; and the description of the results was deleted (Line 321).

  1. The statistics about the melanin quantification should be improved, provide information on the number of independent experiments etc.

Answer: Here both WT and the mutant contained three replicate which was cultured independently, and then the melanin content was measured. The information was added into the experiment procedure description (Line 181).

  1. I would like to see a more complete integration of melanin quantification with the expression data, eg mention that higher expression of PniiA-hsf1 goes together with increased melanisation on nitrate medium

Answer: Integrated discussion of melanin quantification and gene expression was added (Line 447-452): “The colony growth analysis showed that induction or repression of transcription of Cghsf1 did not impair the vegetative growth but significantly influenced the melanism in the PniiA-Cghsf1 mutant. When the expression of Cghsf1 was induced by nitrate, the colony melanism was observational increased in the mutant compared with WT; while when Cghsf1 was repressed by glutamine, the colony melanism was significantly reduced in the mutant, which was in accordance with the melanin quantitation results.”

  • The model of HSF1-CMR1-HSE and final function is also helpful but should be presented as being only part of all regulation going.
  1. For reference, it is much easier to interpret supp Table 2 if you include a line with the canonical HSE sequence in the same table so people can see how similar or different each candidate HSE is

Answer: In supp Table 2, the HSE sequences of each candidate genes (GAATTC, GAAnTTC, and GAAnnTTC) were listed and the conserved sequences were marked with red font. Besides, in supp Table 3, the HSE sequence and the mutated HSE sequence of the probes were marked in red font.

  1. Mention your interpretation of IP/input (%), how should readers read the graphs in fig8, and what are the implications of a certain anti-FLG vs IgG2A ratio. Explain how the tubulin control supports the interpretation of the data as specific binding at 5 other regions.

Answer: Description of the IP/input (%) and the results of the tubulin control was added (Line 357-365): “The chromatin pellet was immunoprecipitated using Anti-FLAG, and the enrichment of promoter fragments containing HSEs was expressed as the percentage relative to the input DNA. The results showed an enrichment in the promoter regions of 4 genes, CMR1, YG, T4HRa, and SCD, compared with when nonspecific antibodies (IgG2a) were used (Figure 8). And the control experiment with the promoter region of β2-tubulin showed that there is no non-specific enrichment for the sequences that do not contain HSE elements.”

  1. Discuss more clearly that binding of hsf1 to HSE is not a pre-requisite for differential gene-expression, eg no binding in T4HRb, T3HR, Laccase (FIG8) but still differential expression for T4HRb and Laccase (FIG7). I know you kind of implied that in other fungi regulation of melanin genes goes through CMR1 but could explain that option in Cg more clearly in the text.

Answer: Thanks for the suggestion. Discussion of the results of gene expression, ChIP-qPCR, and the model of HSF1-CMR1-HSE were added (Line 472-475): “Although the other 4 genes, PKS, T4HRb, T3HR, and Laccase were not identified as the direct targets of CgHSF1, their transcriptions were also under the regulation of CgHSF1.We speculated that it is because that these 4 genes are the direct targets of CMR1 [52, 53], and CgHSF1 could regulate their transcription in an indirect way.”

Reviewer 3 Report

The additional experiments carried out by the authors, as suggested in my former review, show that the promoter of PniiA is induced by inorganic nitrogen and repressed by glutamine. The results shown in Fig. 6 (new) demonstrate that the fungus is able to grow either with induced o repressed expression of Cghsf1. Therefore, the reduction of virulence seen in planta is not caused by the abolishment of growth. The authors demonstrate that the responsible mechanism must be melanization, as the mutant PniiA-Cghsf1, when the promoter is not induced,  shows a clear reduction in the production of melanin and several genes involved in melanin biosynthesis show a drastic reduction in the expression levels.

Author Response

The additional experiments carried out by the authors, as suggested in my former review, show that the promoter of PniiA is induced by inorganic nitrogen and repressed by glutamine. The results shown in Fig. 6 (new) demonstrate that the fungus is able to grow either with induced o repressed expression of Cghsf1. Therefore, the reduction of virulence seen in planta is not caused by the abolishment of growth. The authors demonstrate that the responsible mechanism must be melanization, as the mutant PniiA-Cghsf1, when the promoter is not induced,  shows a clear reduction in the production of melanin and several genes involved in melanin biosynthesis show a drastic reduction in the expression levels.

Answer: Thanks again for the reviewer’s constructive suggestion on the repression of niiA via using glutamine.